# Major distribution shifts are projected for key rangeland grasses under a high-emission scenario in East Africa at the end of the 21st century

Martina Messmer [1,2,9] ✉, Sandra Eckert [3,4], Amor Torre-Marin Rando[5], Mark Snethlage[5], Santos J. González-Rojí [1,2,10], Kaspar Hurni[3], Urs Beyerle [6], Andreas Hemp [7], Staline Kibet[8] & Thomas F. Stocker[1,2]

Grassland landscapes are important ecosystems in East Africa, providing habitat and grazing grounds for wildlife and livestock and supporting pastoralism, an essential part of the agricultural sector. Since future grassland availability directly affects the future mobility needs of pastoralists and wildlife, we aim to model changes in the distribution of key grassland species under climate change. Here we combine a global and regional climate model with a machine learning-based species distribution model to understand the impact of regional climate change on different key grass species. The application of a dynamical downscaling step allows us to capture the fine-scale effects of the region's complex climate, its variability and future changes. We show that the co-occurrence of the analysed grass species is reduced in large parts of eastern Africa, and particularly in the Turkana region, under the high-emission RCP8.5 scenario for the last 30 years of the 21st century. Our results suggest that future climate change will alter the natural resource base, with potentially negative impacts on pastoralism and wildlife in East Africa.

Grasslands are an important ecosystem of East Africa. For example, in Kenya shrublands and grasslands account for the largest share of land cover with 53% and 29% of the total area, respectively[1]. Grasslands provide habitat and grazing grounds for wildlife, which supports tourism, and forage for livestock that ensures the livelihoods of pastoralists and agro-pastoralists. Pastoralism is the primary occupation of 40–45% of the population while for many others it is an important secondary source of income[2]. A key aspect of the pastoral livelihoods is their mobility, which allows them to deal with the climate variability of arid and semi-arid landscapes by following the rains and green pastures[3]. In northern Kenya this mobility is more and more constrained by infrastructure mega-projects and land claims in various forms[4]. For example, land fragmentation and access restrictions pose problems to the mobility of pastoralists and wildlife by threatening access to green pastures.

As the foundation species of grasslands and savannahs, grasses play an essential role in providing structure and function to these ecosystems supporting biodiversity, and delivering crucial ecosystem services such as food production, pollination, climate regulation (including carbon sequestration), and hydrological services, while also being biomes of high conservation value[5–7]. In this study, we focus on seven native key grass species pivotal to pastoralists and wildlife in East Africa: *Cenchrus ciliaris, Cynodon dactylon, Cenchrus mezianus, Cynodon plectostachyus, Digitaria macroblephara, Digitaria milanjiana* and *Themeda triandra*. We selected species based on their utility in pastures and their ecological roles. All the selected

[1]Climate and Environmental Physics, University of Bern, Sidlerstrasse 5, Bern, 3012 Bern, Switzerland. [2]Oeschger Centre for Climate Change Research, University of Bern, Sidlerstrasse 5, Bern, 3012 Bern, Switzerland. [3]Centre for Development and Environment, University of Bern, Mittelstrasse 43, Bern, 3012 Bern, Switzerland. [4]Institute of Geography, University of Bern, Hallerstrasse 12, Bern, 3012 Bern, Switzerland. [5]Institute of Plant Sciences, University of Bern, Altenbergrain 21, Bern, 3012 Bern, Switzerland. [6]Institute for Atmospheric and Climate Science, ETH Zurich, Universitätstrasse 16, Zurich, 8092 Zurich, Switzerland. [7]Department of Plant Systematics, University of Bayreuth, Universitätsstrasse 30, 95440 Bayreuth, Germany. [8]Land Resource Management and Agricultural Technology Department, Faculty of Agriculture, University of Nairobi, P.O Box 29053-00625 Nairobi, Kenya. [9]Present address: Geoscience and Remote Sensing, Faculty of Civil Engineering and Geosciences, Delft University of Technology, Stevinweg 1, 2628 CN Delft, The Netherlands. [10]Present address: Department of Physics, University of the Basque Country, Barrio Sarriena s/n, 48940 Leioa, Spain. ✉e-mail: m.messmer@tudelft.nl

species are perennial plants and well adapted to arid and semi-arid eco-systems. The seven species occurred in about 60% of 659 plots of grasslands and dry savannah woodlands in East Africa[8]. Furthermore, these grasses represent the major structural inflorescence types of grasses: spikes and spike-like panicles, panicles and (digitate) racemes. The majority of the species have tufted and/or stoloniferous morphology and as such play an important role in the fire ecology, a key determinant of grasslands, soil hydrological properties and in providing much needed soil cover, facil-itating water infiltration into the soil column and subsequently reducing run-off that causes soil erosion[9–11].

*C. ciliaris* is known for its adaptability to arid and semi-arid regions as it is very drought tolerant and responds quickly to rain by growing[12]. Due to its drought tolerance, it is put under cultivation for grazing and used for reseeding degraded rangelands[13,14]. *C. dactylon* is valued not only for lawns and sports fields but also for pastures, owing to its trampling resistance and versatility across various environments[15,16]. *Cenchrus ciliaris* and *Cynodon dactylon* are the most preferred dry season pastures as they tolerate drought conditions and therefore are widely available[17]. *C. mezianus* is also a much appreciated species for dry season pastures in Kenya[17] and a geographical specialist with a restricted distribution, emphasizing its unique ecological niche in parts of East Africa. *C. plectostachyus* is very competitive and grazing tolerant, and its fitness increases with the activity of large herbivores. Additionally, it is an exceptionally nutritious grass compared to other co-occurring grass species[18]. *D. macroblephara* is a dominant grass species in the Serengeti National Park and is widely distributed in East Africa[19]. *D. macroblephara* shows an increase in abundance under frequent and intense grazing[20]. Moreover, *D. macroblephara* ranked highly by pastoral com-munities due to its high digestibility and ability to increase milk yield and weight of livestock[14,17]. *D. milanjiana* is a highly variable grass species and adaptive to changes in rainfall and temperature[21]. Finally, *T. triandra* represents a native and ecologically important grass essential for main-taining biodiversity and ecological health. *T. triandra* is relatively palatable, but continuous or uninterrupted grazing, can lead to a decline in its abundance[10]. Such a decline is often related to a decline in grazing value, species richness, species cover and therefore, ecosystem function[10]. The selection of these grass species highlights their roles in supporting pastoral livelihoods, ecological balance, and biodiversity conservation. The combi-nation of these selected widespread and important savannah grasses is representative of typical East African savannah grassland ecosystems[7,22].

Future climate change poses a threat to grassland ecosystems, poten-tially altering the availability of suitable areas for wildlife and livestock grazing, and changing the seasonal mobility needs of pastoralists and wildlife. Modelling the future impacts of climate on grasslands is essential to provide strategic foresight into the future availability of grasslands and the changing mobility needs of livestock and wildlife. Despite the importance of these changing needs, our understanding of how climate change will affect these grass species remains limited. To date, only one study has examined the impact of climate change on *C. ciliaris* in southern North America, where it is considered a highly invasive species[23]. This study found that changes in temperature and precipitation largely explain the spatial varia-bility of the species, and suggests that its range is expected to expand westward and southward under future climate scenarios[23]. However, there is a notable lack of similar assessments for East Africa, despite the important insights they could provide for conservationists, pastoralists, and the agri-cultural sector. Such information is crucial for identifying potential conflicts and developing conservation strategies ahead.

Currently, our understanding of how climate change is affecting grass species and the overall composition of grasslands in East Africa is incom-plete. One challenge is the lack of maps and records that track the location and evolution of grasslands over time[24,25]. In addition, the use of global climate model (GCM) output in species distribution models (SDMs) introduces errors due to the coarse resolution of these models, which often fail to accurately capture the regional impacts of climate change in East Africa. As a result, the effects of climate change on grass species habitats and grassland composition are not well understood and ambiguous[26], nor are the

effects of changing livestock grazing patterns on grassland ecosystem dynamics[27,28]. Addressing these challenges is essential to better understand the impacts of anthropogenic climate change on grassland ecosystems in East Africa and to design appropriate adaptation and mitigation strategies.

This study aims to address these gaps in our understanding of East African grasslands under climate change. We aim to determine how climate change could alter grassland habitats, change the composition of key grass species in grasslands, and highlight the potential impacts of these changes on (agro-)pastoralists, their livestock and wildlife. We also aim to provide evidence to guide the design of effective conservation areas. To achieve these goals, we generate detailed geospatial projections of the current and future distribution and co-occurrence of seven key grass species under a high-emission scenario by the end of the 21st century. We use a regional climate model to produce fine-scale projections, which are essential to capture the complex and diverse climate of the region. Using the Weather Research and Forecasting (WRF) model[29] at 9 km horizontal resolution, driven by the Community Earth System Model (CESM)[30] under the high-emission sce-nario RCP8.5[31], we obtain the necessary climate data for the current period and for the end of the century. These data, combined with other non-climatic variables, will be used to train species distribution models (SDMs) using machine learning algorithms. Using real presence and absence data for seven key grass species (Supplementary Table S1), the models project current and future occurrences, providing important insights into the future of East African grasslands. This is because the combination of the selected widespread and important savannah grasses provide a meaningful indicator of typical East African savannah grassland ecosystems[7,22].

## Results
### Changes in bioclimatic predictors under the high-emission scenario RCP8.5

In East Africa, the mean annual temperature (Bio1) is projected to increase by 2–3.5 °C by the end of the 21st century compared to the present (Fig. 1a), with the strongest temperature rise projected for the east Kenyan plains (A1; see Supplementary Fig. S1 for the location of the different regions A to C and Section "Study region") and the Ethiopian highlands (B4). Conversely, the Turkana region (A2) and the northern part of the Kenyan highlands (B1) show the least warming. Two variables describe temperature variability: the mean diurnal range (Bio2) and temperature seasonality (Bio4). The mean diurnal range is projected to decrease markedly under the high-emission scenario, with a decrease of 2–3 °C corresponding to a reduction of around 15–20% of the present-day diurnal range (Fig. 1b). This decrease is most pronounced in the Lake Turkana region and the northern part of both the Kenyan highlands and the east Kenyan plains, resulting from a disproportional increase of the daily minimum temperature. The mean annual temperature season-ality is projected to increase strongly in the Turkana region and parts of the Kenyan highlands, while decreasing in the Serengeti plains (B2), Tanzanian highlands (B3) and Kitui county (A4, Fig. 1c).

The reduced warming in the Turkana region coincides with a strong increase in mean annual precipitation (Bio12) of 400–600 mm per year (Fig. 1d). The greatest increase in precipitation, up to 800 mm per year, is projected for the centre of the Kenyan highlands. In large parts of the east Kenyan arid lowlands, the increases are around 200 mm per year. A decrease in precipitation is projected only for the south-eastern part of the study area, including the Ser-engeti plains and the Tanzanian highlands. In addition to changes in total annual precipitation, seasonal precipitation patterns will also change, with precipitation in the driest month (Bio14) projected to increase in the high-elevation areas of the Kenyan highlands and along the coast (C), amounting to a doubling and tripling of present amounts in the two regions, respectively (Fig. 1e). Conversely, the model projects a decrease of up to 10 mm per month in the northern part of the Serengeti plains, which means that during the driest month of the year, part of the region receives no rainfall at all.

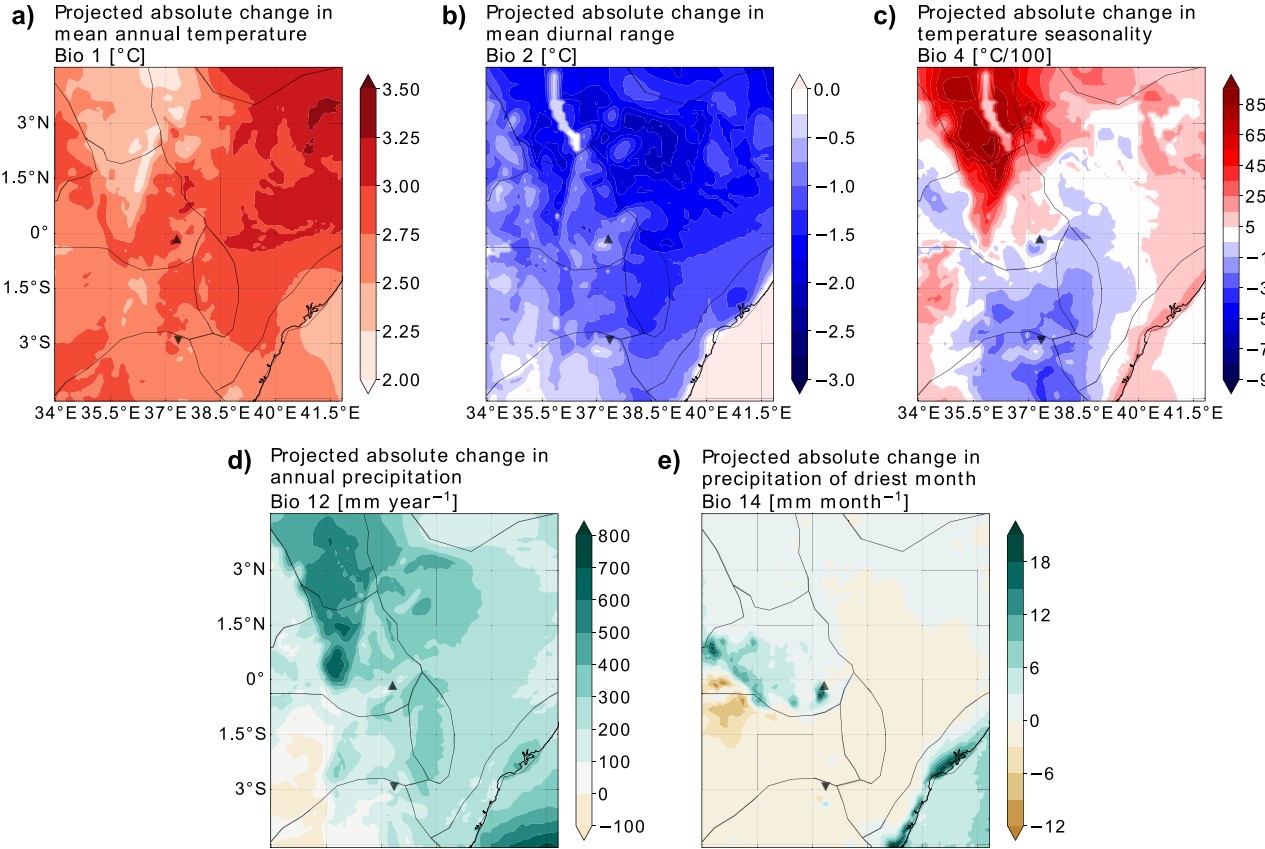

**Fig. 1 | Changes in bioclimatic variables between the present and the future.**
Changes between the future and the present (future, i.e. RCP8.5 at 2071–2100, minus present, i.e. 1981–2010) for **a** mean annual temperature (Bio 1) in °C, **b** mean annual diurnal temperature range (Bio 2) in °C, **c** mean annual temperature seasonality (Bio 4) in °C/100, **d** mean annual precipitation sum (Bio 12) in millimetres per year, and

**e** mean precipitation sum in the driest month (Bio 14) in millimetres per month. The grey boundary lines indicate physiographic units of East Africa (see Methods and Supplementary Fig. S1). The upward-pointing black triangle indicates the location of Mount Kenya, and the downward-pointing one marks Mount Kilimanjaro.

## Patterns and trends for seven key grass species

For all studied key grass species, tree cover (TC) emerged as the most critical predictor (Table 1, RF model). Notably, TC is the most crucial predictor for *C. ciliaris* (BRT and RF model), *D. milanjiana* (BRT and RF model), *C. mezianus* (BRT model), and *T. triandra* (BRT and RF model).

Temperature-related factors (Bio1, Bio2, or Bio4) ranked as the second most crucial predictor across all seven species, except for *C. plectostachyus*, which is most sensitive to precipitation during the driest month. Overall, all bioclimatic predictors are important for the key grass species, with *C. plectostachyus* depending mainly on precipitation while the others on temperature. Soil texture was the least important predictor for all species, with the human footprint index (HFI) and the cation exchange capacity (CEC) playing a minor role. The BRT model confirms these findings, with only minor shifts in the importance rankings when compared to the RF model (Supplementary Table S2).

We now consider the response to the high-emission scenario RCP8.5[31]. Only two of the key grass species, *C. ciliaris* and *D. milanjiana*, show a consistent overall range expansion according to the spatial patterns of species occurrence in both SDMs (Fig. 2a, f). *C. ciliaris* is projected to increase its range by 4.6% (RF) to 5% (BRT), which corresponds to an increase of 29,600 km² (RF) to 30,700 km² (BRT) (Table 2 and Supplementary Table S3). Similarly, *D. milanjiana* is projected to increase its range by 2.7% (RF) to 17.1% (BRT), amounting to an increase of 12,200 km² (RF) to 79,800 km² (BRT). For all other species, the overall range is expected to shrink in the future, except for *D. macroblephara* and *T. triandra*, which show inconsistent trends between the two models. Among the key grass species, *C. mezianus* is projected to have the strongest consistent decrease

under future climate conditions, with a decrease ranging from 34.6% to 35.8%, depending on the model. The RF model also projects a similarly strong decrease for *C. plectostachyus*, amounting to 44.4% or 245,600 km², which is not confirmed by the BRT model, projecting a decrease of only around 10.8% or 64,000 km².

The results of the SDM indicate the absence of *C. ciliaris* in large parts of the Kenyan highlands, but overall an expansion of this species by 47,100 km² is projected, mostly in elevated areas in the north Ugandan plains (A3), the Serengeti plains, and around Mount Kilimanjaro, as shown in Table 2 and the green areas in Fig. 2a and Supplementary Fig. S2a. The consistent change between the two models in *T. triandra* (Fig. 2b and Supplementary Fig. S2b) is the range contraction around Lake Turkana, in the western part of the Serengeti plains, and in the Tanzanian highlands. In the east Kenyan plains and in the Tanzanian highlands, *T. triandra* is projected to slightly expand its range. This expansion is larger in the BRT model (Supplementary Fig. S2b) than in the RF model (Fig. 2b). As a result, the overall change ranges widely, from an increase of 11,200 km² in the BRT model to a decrease of 76,300 km² in the RF model (Table 2 and Supplementary Table S3). For *C. dactylon* both the RF and BRT models indicate a range contraction in the east Kenyan plains, Kitui county, northern part of Lake Turkana and in the northern part of the Kenyan highlands (Fig. 2c and Supplementary Fig. S2c). The RF model projects an overall range contraction of 91,900 km², while the BRT model projects a slightly smaller decrease of 68,300 km². Simultaneously, both models suggest a range expansion of *C. dactylon* in the northern Kenyan highlands and in the Tanzanian highlands. *C. plectostachyus* is projected to almost completely disappear in the Lake Turkana region (Fig. 2d and Supplementary Fig. S2d). At the same time,

**Table 1 | RF model variable importance**

| | C. ciliaris | | C. dactylon | | C. plectostachyus | | D. macroblephara | | D. milanjiana | | C. mezianus | | T. triandra | | Overall | |
|---|---|---|---|---|---|---|---|---|---|---|---|---|---|---|---|---|
| | TSS | 0.63 | TSS | 0.58 | TSS | 0.75 | TSS | 0.66 | TSS | 0.52 | TSS | 0.7 | TSS | 0.72 | TSS | |
| | TC | 19.50 | Bio2 | 16.80 | Bio14 | 25.00 | Bio2 | 21.40 | TC | 12.70 | Bio2 | 18.1 | TC | 13.60 | TC | 14.3 |
| | Bio1 | 12.70 | Bio12 | 10.60 | TC | 15.40 | TC | 15.80 | Bio1 | 12.60 | TC | 15.4 | Bio4 | 12.40 | Bio2 | 13.8 |
| | slope | 11.70 | Bio4 | 10.40 | CEC | 9.00 | Bio4 | 11.60 | Bio2 | 9.90 | Bio14 | 12.8 | Bio2 | 11.30 | Bio14 | 11.8 |
| | Bio14 | 11.30 | Bio14 | 9.50 | Bio1 | 8.90 | slope | 10.50 | slope | 9.90 | slope | 12.5 | Bio1 | 10.60 | slope | 10.0 |
| | Bio2 | 11.30 | Bio1 | 9.40 | Bio4 | 8.80 | d2ww | 9.40 | Bio4 | 9.70 | Bio4 | 9.2 | Bio14 | 9.60 | Bio1 | 9.9 |
| | Bio4 | 7.30 | HFI | 9.40 | Bio12 | 8.50 | Bio1 | 8.00 | Bio14 | 9.40 | Bio1 | 7.1 | Bio12 | 9.10 | Bio4 | 9.9 |
| | Bio12 | 7.10 | slope | 9.00 | Bio2 | 7.70 | HFI | 7.00 | Bio12 | 8.90 | Bio12 | 6.7 | slope | 8.60 | Bio12 | 8.0 |
| | d2ww | 6.90 | d2ww | 8.00 | slope | 7.50 | Bio12 | 5.20 | HFI | 8.70 | HFI | 5.8 | CEC | 7.70 | d2ww | 7.1 |
| | HFI | 5.00 | TC | 7.90 | d2ww | 4.80 | Bio14 | 4.90 | d2ww | 8.00 | d2ww | 5.3 | d2ww | 7.20 | HFI | 6.5 |
| | CEC | 5.00 | CEC | 6.90 | HFI | 3.30 | CEC | 4.10 | CEC | 7.60 | CEC | 4.8 | HFI | 6.60 | CEC | 6.4 |
| | Texture | 2.20 | Texture | 2.10 | Texture | 1.10 | Texture | 2.10 | Texture | 2.60 | Texture | 2.4 | Texture | 3.40 | Texture | 2.3 |

Variable importance in descending order for the RF model (for the BRT model, see Supplementary Table S2). The models' True Skill Score (TSS; Methods) are indicated below the names of the grass species (see Supplementary Table S1). The variable importance is given by the IncNode Purity (a higher IncNode Purity indicates a more important variable for separating the data into homogeneous groups), which is normalized by the total sum of the importance over all predictors of each species, and therefore given in percent. The last column shows the average over all species, therefore no TSS is available. All variable abbreviations can be found in Supplementary Table S4.

both models suggest a range expansion in the Serengeti plains. A range contraction is projected for *D. macroblephara* at the boundary of the east Kenyan plains and the Ethiopian highlands, in the Kenyan highlands, and in

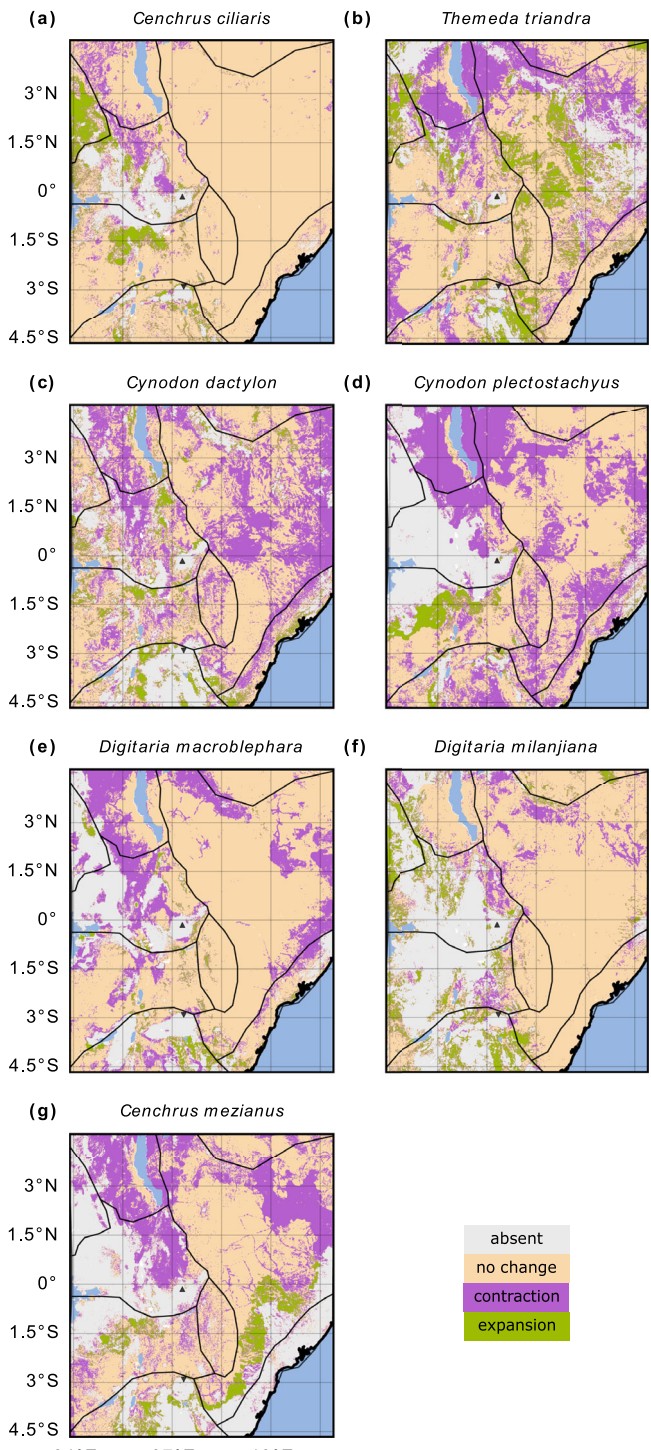

**Fig. 2 | Changes in the occurrence of the seven grass species between the present and the future predicted by the RF model. a** *C. ciliaris*, **b** *T. triandra*, **c** *C. dactylon*, **d** *C. plectostachyus*, **e** *D. macroblephara*, **f** *D. milanjiana*, and **g** *C. mezianus*. The upward-pointing black triangle indicates the location of Mount Kenya, and the downward-pointing one marks Mount Kilimanjaro. The grey, beige, pink and green shaded areas indicate absence, no change, range contraction (species disappear from area) and range expansion (species invade area), respectively, for each species under future climate conditions. For more details see Section "Species data and species distribution models". For the BRT model, see Supplementary Fig. S2.

**Table 2 | Change in area of grass species**

| Species | Absent [100 km²] | No change [100 km²] | Contraction [100 km²] | Expansion [100 km²] | Change [100 km²] | Change [%] |
|---|---|---|---|---|---|---|
| *C. ciliaris* | 991 | 6721 | 176 | 471 | 296 | 4.6 |
| *C. dactylon* | 1811 | 4600 | 1433 | 514 | −919 | −16.6 |
| *C. plectostachyus* | 2215 | 3081 | 2760 | 304 | −2456 | −44.4 |
| *D. macroblephara* | 1729 | 5239 | 1219 | 172 | −1047 | −16.7 |
| *D. milanjiana* | 2696 | 4697 | 422 | 544 | 122 | 2.7 |
| *C. mezianus* | 3060 | 3134 | 1912 | 253 | −1659 | −34.6 |
| *T. triandra* | 2731 | 3306 | 1543 | 780 | −763 | −18.8 |

Area of different grassland types in 100 square kilometres, separated into absent, no change, contraction, expansion, and an overall change, obtained from the RF model (for the BRT model, see Supplementary Table S3). The last column indicates the relative change in per cent of the relevant species' area of presence under present climate conditions.

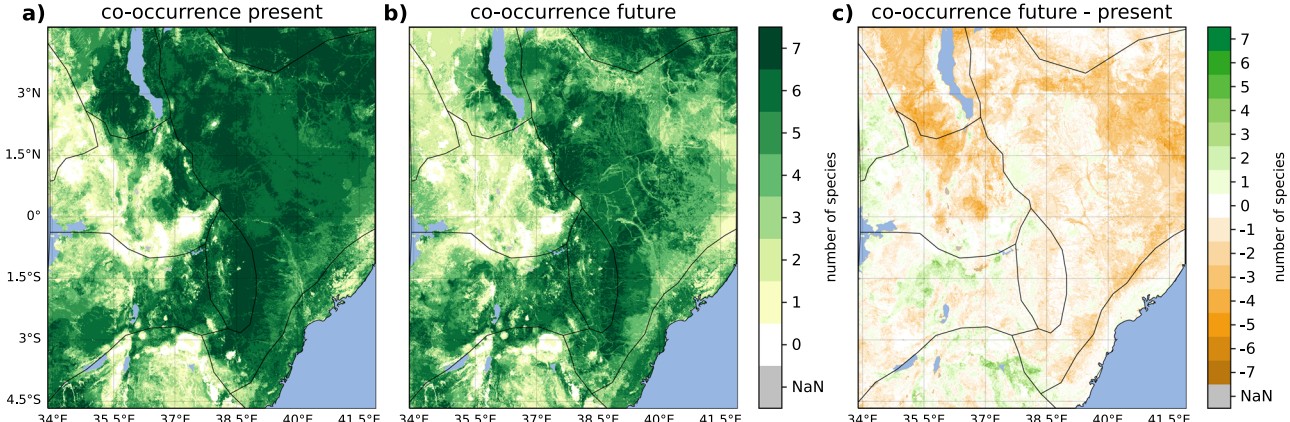

**Fig. 3 | Predicted co-occurrence of the seven grass species.** Predicted co-occurrence of the seven grass species under (**a**) present climate conditions and (**b**) future climate conditions. Panel **c** indicates the change in the co-occurrence of the grass species under future climate (RCP8.5, 2071–2100) compared to present climate (1981–2010) conditions obtained from the RF model (for the BRT model, see Supplementary Fig. S3). Orange (green) shadings in the panel **c**) indicate a decline (increase) in the number of species that populate the same area.

the Serengeti plains (Fig. 2e and Supplementary Fig. S2e). Other pattern changes for *D. macroblephara* are difficult to interpret, as they are patchy and inconsistent with the BRT model results. An expansion is projected for *D. milanjiana* (Fig. 2f and Supplementary Fig. S2f) in the north Ugandan plains, the Kenyan highlands, the Serengeti plains, and the Tanzanian highlands. Generally, the two models agree on these regions, but the BRT model projects a much stronger increase than the RF model. Finally, both SDMs project a range contraction of *C. mezianus* in the surroundings of Lake Turkana, the north-eastern Kenyan highlands and in the eastern part of the east Kenyan plains. Simultaneously, a range expansion in the southern part of the east Kenyan plains is projected (Fig. 2g and Supplementary Fig. S2g).

To summarize the changes of all seven grass species, their co-occurrence is investigated under present and future climate (Fig. 3 and Supplementary Fig. S3 for the RF and the BRT models, respectively). The co-occurrence is expressed in the number of individual grass species present at a given grid point. Our study region shows a separation into a part with a projected increase in grass species co-occurrence which is in the western part, i.e., north Uganda plains, western part of Kenya highlands, the Serengeti plains, and the Tanzanian highlands. The other part of the study area is projected to see a decline in the co-occurrence of grass species. A particularly pronounced decrease is projected for the greater Turkana region, extending into the Mount Kenya region. In addition, most of arid eastern Kenya plains are likewise projected to see a future decline in species co-occurrence by different grass species. It should be noted, however, that the coverage of the actual species presence and absence data is very sparse in the eastern part of the domain, so these results

must be treated with caution. The changes in co-occurrence under global warming are very similar for the RF and the BRT models in most regions (Fig. 3 and Supplementary Fig. S3).

## Discussion

Our results based on dynamical downscaling highlight the potential impacts of climate change on grass species occurrence in the region. With an average projected warming of 2–3.5 °C for the last 30 years of the 21st century in a high-emission scenario (RCP8.5)[31], we are within the range of CMIP5 models, but in the cold quartile of the model spread[32]. The wetting of the region, which our model suggests, is in line with other climate change projections for East Africa[33,34]. Since we are using only one climate simulation, it is important to consider alongside other existing climate change projections. A comparison with the widely used WorldClim CMIP5 data sets shows that the results of our dynamical downscaling are within the uncertainty range of the WorldClim ensemble members with respect to temperature and precipitation, but with more realistic and finer-scale patterns (see Section "Model verification: future climate response over East Africa under RCP8.5"). Additionally, the dynamical downscaling allows to approximately maintain the physical consistency between the bioclimatic variables. When considering climate change, SDMs can contribute to an improved understanding of future habitat suitability for a given species and species range expansion and contraction. When using SDMs to understand climate-change-related habitat changes of plant species within the same ecosystem and geographic region, it is particularly important that climate data and future climate projections are as accurate as

possible and provide sufficient spatial detail to capture subnational to local habitat changes[35,36].

Both SDMs that we applied to the data revealed tree cover as the most critical predictor across all key grass species. This is probably because these grasses struggle to grow in areas with high tree density due to limited sunlight. With an increase in annual precipitation but potentially also elevated atmospheric $CO_2$, the transition of grasslands into woodland in savannah ecosystems will further be promoted[37]. Our study has shown that climate related variables such as temperature and precipitation are important for habitat suitability of different grass species. The distribution of species suitability is primarily shaped by regional precipitation patterns, while temperature largely determines the maximum elevation that grass species can inhabit, as higher elevations tend to have lower temperatures. Additionally, soil properties influence the more detailed, localized distribution of grass species[10,19].

Before we discuss the changes in grass species suitability under global warming, we must acknowledge that statistical species distribution models are inherently designed to identify nonlinear connections, which limits their ability to establish causal relationships. Hence, some changes in the patterns of grass species can be attributed directly to changes in bioclimatic variables, which are the only variables that differ between the two periods under consideration. However, most of the changes observed are the result of a nonlinear interplay between various variables, and the exact reason for the change cannot be determined. This is, for example, the case for C. dactylon, whose changes are related to a complex interplay between the 11 selected variables and the changes in some of these variables.

The results of the two SDMs indicate the absence of C. ciliaris in large parts of the Kenyan highlands, which is not surprising given the species' preference for low rainfall and high temperatures[23]. C. ciliaris is known for its ability to withstand and recover from droughts better than other grass species[38]. As such, this species is expected to expand its range towards areas with drier and warmer conditions due to climate change[23]. C. ciliaris is a highly valued pasture grass in the tropics and is known for reducing the abundance of unpalatable species due to its dominance and ability to produce biomass quickly in response to unpredictable precipitation[39,40]. With climate change, higher-elevation areas in East Africa are expected to become suitable for C. ciliaris, which is what the models project for the elevated regions in the western part of the domain, making it more available for pastoral use.

The main change in T. triandra's distribution – which is captured by both the RF and the BRT models – is the range contraction around Lake Turkana. For this region, the climate model projects a substantial increase in temperature seasonality. Areas experiencing a range contraction of T. triandra may experience a decline in grassland ecosystem services for pastoralists in the future. This is because T. triandra is an important food source for both livestock and wildlife[10,41] and plays a central role in the ecological dynamics of savannahs. The species contributes substantially to pasture resilience and its decline is associated with a deterioration of grazing value[10,41,42]. Additionally, a decline in T. triandra is associated with a decline in species richness and ecosystem function and therefore, impacts biodiversity, not only of grass species, but also of herbivores[10]. In regions where the temperature seasonality remains the same or decreases under the future climate, such as in the east Kenyan plains and in the Tanzanian highlands, T. triandra is projected to slightly expand its range.

Both models project a decline in the habitat suitability of C. dactylon. The areas where a range contraction is predicted could become less suitable for feeding livestock and wildlife, as C. dactylon is a valuable food source. Particularly in the dry season, C. dactylon is a valuable grazing forage to improve livestock production, so a range contraction could predispose livestock and wildlife to a sub-optimal performance. Additionally, the species is excellent in controlling soil erosion[17,43], which would be a highly preferred characteristic due to the projected strong increase in precipitation amounts under the high-emission scenario RCP8.5.

The decline of C. plectostachyus around Lake Turkana is likely due to increased precipitation and temperature seasonality, which will make the future climate in that region unsuitable for this species. C. plectostachyus is palatable to livestock and is valuable as fodder[40], given its high nutritious value[18], so its decrease might also have a negative impact on pasture quality in the affected areas.

The reduction of D. macroblephara near the boundary of the east Kenyan plains and the Ethiopian highlands is probably due to a slight decrease in diurnal temperature range, led by an increase in the minimal temperature. Other pattern changes are challenging to interpret, as they are patchy and sometimes both model results are inconsistent.

The increase in D. milanjiana is probably related to the rise in temperature in higher-elevation areas, such as at the boundary of the north Ugandan plains, the Lake Turkana region, and the Tanzanian highlands. With climate change, higher-elevation areas are expected to become suitable for D. milanjiana, making it more available for pastoral use.

Finally, the changes in the habitat of C. mezianus cannot be directly related to projected climatic changes, because of the nonlinear interplay between the different variables. Nevertheless, the implications of the projected decrease for C. mezianus are highly relevant, as this species is valuable for livestock and wildlife during dry periods. Despite being relatively unpalatable when mature[44,45], C. mezianus is consumed by livestock and wildlife during periods of feed deficit[46], so its decrease might negatively impact the availability of pasture resources during harsh conditions.

Taken together, the changes of the individual grass species could serve as a first-order measure of grassland biodiversity and its projected change. The co-occurrence of different grass species is projected to decrease in the future, which implies reduced grass species diversity by the end of the 21st century. This indicates a first risk of grassland biodiversity loss in this region under a high-emission scenario. The decrease in co-occurrence is particularly pronounced in the greater Turkana region, extending partly into the Mount Kenya region. Particularly in the Turkana region, most of the investigated grass species will not find suitable conditions anymore under global warming. This region is projected to experience strong climatic changes, including a substantial increase in annual precipitation sums and in the annual temperature seasonality. The combination of decreased grass species suitability of almost all investigated species and the strong increase in precipitation and its extremes might increase the threat of floods, soil erosion and landslides in that area. In addition, most of the arid eastern Kenya plains are likewise projected to see a future decline in species co-occurrence by several grass species. It should be noted, however, that the coverage of the actual species presence and absence data is very sparse in the eastern part of the domain, so these results must be treated with caution. The projected range contraction of C. mezianus and C. dactylon, which today provide an important source of fodder in seasons when conditions get particularly harsh, and T. triandra, which is an indicator of healthy grassland ecosystems, is a source of concern. The projected changes, both the increase and decrease in co-occurrence, impact the local pasture availability and composition under climate change and therefore, might also change the wildlife and pastoral mobility. This has the potential for new conflicts over access to, and availability of, essential resources.

Even though the goal of our analysis is to investigate the impact of climate change on the different grass species, we would like to point out that climate variables are not the only, and perhaps, not the main factors that are responsible for changes in grass species. With regard to savannahs, anthropogenic climate change, and particularly the projected increase in precipitation, could favour the spread of woody plants[47] and therefore, indirectly put further pressure on grassland species. Human disturbance and land degradation due to overgrazing could further lead to a decline in various grass species[17]. The combination of land fragmentation (e.g., due to infrastructure, cropland expansion, protection, conservation, economic investments, or increased population), and the projected changes in the habitats of the investigated grass species may pose equally serious problems for pastoralists and native wildlife. Additionally, habitat changes and disturbances might enable invasive species to fill the emerging gaps and further expand their habitats. This could exacerbate the ongoing replacement of valuable native grass species by various invasive species (e.g., *Parthenium hysterophorus*,

*Xanthium strumarium, Prosopis juliflora, Lantana camara*) and therefore, further threaten the native biodiversity. It is known that such invasions are facilitated by land degradation due to overgrazing and deforestation, as well as by climate change[48], although in a meta-analysis it was found that in some areas of the world, land management effects outweigh the influence of climate change and increasing $CO_2$ concentrations on grassland dynamics[49].

## Conclusion

Under present-day climate conditions, the arid lowlands of eastern and northern Kenya seem favourable to all studied grassland species. However, future climate change under the high-emission scenario RCP8.5 is expected to alter the distribution and composition of grassland ecosystems. While *C. ciliaris* and *D. milanjiana*, show a slight overall increase in habitat suitability, species such as *C. dactylon*, *C. plectostachyus* and *C. mezianus* are projected to experience notable range contractions. The Turkana region, in particular, is expected to be severely impacted, with a near-complete absence of the studied species under the high-emission scenario. These negative effects are likely driven by increased precipitation and seasonal temperature, which create unfavourable conditions for many grass species. Elevated regions present less favourable conditions for some of the considered species under present-day climate conditions. However, the projected higher temperatures will possibly help some of the grasses to conquer these regions. With this study we tried to anticipate the currently still uncertain changes in grass species, key for wildlife and livestock of pastoralists, under climate change. Our results are valuable for assessing the economic potential of the region and the sustainable long-term planning, for example when designing livestock and wildlife corridors or highway crossings. While our analysis focused on climate change impacts on bioclimatic variables, future studies should consider additional factors such as vegetation changes, land use, and human activities, which could further influence grassland conditions. Additionally, exploring the competitive dynamics among grass species and potential benefits to other species not included in this study would provide a more comprehensive understanding of grassland evolution under future climate scenarios. It should be noted that the results presented here provide potential areas where grass species could grow, but our study does not consider the human influence of grass cultivation or reseeding of degraded rangelands, as has occurred in the recent past[13,14].

## Methods
### Study region

The study region comprises East Africa, with a focus on Kenya including bordering areas in Ethiopia and Somalia. To facilitate the discussion of the climate and species distribution patterns, we divide the study region into physiographic units (Supplementary Fig. S1), loosely based on the agro-ecological zones of Kenya[50]. Agro-ecological zones categorize a larger land area into smaller sections that share similar traits regarding land suitability, potential production, and environmental impact[50]. The nine physiographic units pertain to three main regions:

- The hot and arid lowlands of the north and east (A, greenish areas in Supplementary Fig. S1):
    - A1: East Kenyan plains (from Marsabit and Mandera in the north to Tana River County in the south), including adjacent southern Somalia
    - A2: Lake Turkana: Turkana County and north-western Marsabit
    - A3: North Ugandan plains (Kaabong and Moroto Districts)
    - A4: Kitui County
- The eastern rift highlands and plateaus (B, reddish areas in Supplementary Fig. S1):
    - B1: Kenyan (and Ugandan) highlands to Lake Victoria
    - B2: Serengeti Plains (south-west Kenya and northern Tanzania)
    - B3: Tanzanian highlands (Ngorongoro, Meru, Kilimanjaro, Eastern Arc Mountains)
    - B4: Ethiopian highlands (southward extension of the Ethiopian highlands)
- The coast (C, purple area in Supplementary Fig. S1)

### Soil and landscape data

For the species distribution modelling, we use 18 environmental variables that characterize the study area (Supplementary Table S4, right column). We use two topographic variables, elevation and slope, derived from the ALOS World 3D (AW3D) digital elevation model provided by the Japan Aerospace Exploration Agency (JAXA)[51]. We also include two variables related to surface water, namely Euclidean distance to rivers and water bodies, respectively, which we generated ourselves. To account for tree cover and photosynthetic active vegetation, we use Hansen's global tree cover data set of 2010[52] and a normalized vegetation index (NDVI) generated from a cloud-free Landsat-8 dry-season satellite data mosaic. For the characterization of the soils, we use the SoilGrids data sets provided by ISRIC[53]. The following variables are considered from soil depth to bedrock: per cent sand, clay, and silt; soil texture; cation exchange capacity (CEC); acidity ($pH_{H_2O}$); nitrogen (N); soil organic carbon (SOC); salinity; soil depth; and soil class. The Afrisoils data set is used for Aluminium exchange capacity[54]. We include anthropogenic influences in our model by means of the global human footprint index (HFI) provided by the Wildlife Conservation Society (WCS)[55]. The spatial resolution of these raster data sets varies between 30 m (topographic as well as water, vegetation, and tree cover data), 250 m (soil data) and 1 km (HFI and soil data). Using a bilinear interpolation, we resample them to a common resolution of 1 km.

### Climate models

To quantify climate change, we use a model chain from global to regional scales. The GCM used is the Community Earth System Model (CESM, version 1.04)[30]. It consists of four components for atmosphere, ocean, sea ice, and land, which are fully coupled. The horizontal resolution of the atmosphere and land components is 1.25° × 0.9° (longitude × latitude). The model is used to perform a historical simulation from 1850 to 2005 and a future simulation under the high-emission scenario RCP8.5 from 2006 to 2100[56,57]. In this study, we use two 30-year time slices, 1981–2010 and 2071–2100, to represent the present and a future climate, respectively.

The global model output of the two selected time slices is used to drive the regional climate model, which in this study is the WRF model (version 3.8.1)[29]. In the model setup, we make use of a parent and a nested domain to obtain a fine horizontal resolution over Kenya (Supplementary Fig. S4). The parent domain (D1) has a horizontal grid resolution of 27 km and covers a large area of Africa, from the Sahel area to Madagascar (15° N–20° S, 5° W–70° E). The nested domain (D2) has a grid spacing of 9 km and covers Kenya, extending westward to north-eastern Congo. The vertical extent of the atmosphere reaches up to a level of 50 hPa, separated into 49 eta levels. The regional model solves the fundamental physical equations governing the atmospheric dynamics, e.g., thermodynamics, radiation and moisture transport, on this grid and thus obtains a dynamical downscaling. Sub-grid-scale atmospheric processes, such as convection and microphysical cloud processes, are parameterized using the optimal configuration for Kenya[58]. The output of the model is obtained every hour, thus providing a high spatial and temporal resolution.

To model grass species, the absolute temperature ranges and precipitation amounts must resemble observations. To correct particularly the bias of the global model CESM, the following two-step approach is applied: (1) WRF is used to downscale the latest reanalysis, ERA5, provided by the European Centre for Medium-Range Weather Forecasts (ECMWF)[59]. This simulation is composed of a 20-year climatology for the period of 1999–2018, which provides the baseline for the climate projection. ERA5 provides relatively high-resolution and good-quality data for the region. The chosen configuration results in a good match between precipitation and temperature at weather stations[58]. (2) The monthly climatological differences between the future and present RCM simulations based on CESM are added to this simulation. This so-called delta change method is considered to provide robust climate impact scenarios[60,61] and is therefore widely applied in studies on climate change e.g. refs. 62–64. In other words, the 19 bioclimatic variables (BioClim[65]) are first calculated for all three RCM simulations: the ERA5, the present, and the future climate simulations

(Supplementary Table S5). After that, the future and the present BioClim variables are subtracted from each other and added to the ERA5-based BioClim variables, resulting in an adjusted future climate simulation. The delta change approach is applied to both temperature and precipitation. A correction by a percentage change, which is usually applied to precipitation, was not possible since there are grid points with no rainfall at all. Grid points for which the delta change method results in a negative future precipitation are set to zero. To obtain the same spatial resolution for all predictors, the bioclimatic variables are resampled to a 1 km grid by means of bilinear interpolation.

### Species data and species distribution models

To understand the effect of climate change on the distribution of grasslands in East Africa and particularly Kenya, we assess seven grass species that are relevant as a source of food for livestock and wildlife (Supplementary Table S1). The species are selected in two steps. An initial selection of 17 species is made based on our own extensive literature review. It is then narrowed down to the eight most important species based on expert scientific knowledge. One species is excluded after model evaluations due to a lack of data (see below).

Presence data for the eight grass species are collected from five spatially explicit databases: Global Biodiversity Information Facility (GBIF)[66–72], sPlot – The Global Vegetation Database[73], RAINBIO[74], Vegetation Database East Africa (VDEA)[8], and Swea-dataveg: A vegetation database for Sub-Saharan Africa[75]. In addition to presence data, sPlot and VDEA also provide several real absence data points. The different data sets are combined in a database. All presence points (including historical observations) are retained; absence points are selected through gridded randomization (grid size 5 × 5 km²) retaining one random absence point per grid cell. Absence points within 5 km from a presence point are discarded. The number of these points for each species is provided in Supplementary Table S1 and displayed in Supplementary Fig. S5, while Supplementary Fig. S6 disaggregates the points by repository and species. A high density of presence and absence points is found in the western and southern parts of the study area, whereas the east Kenyan plains are relatively sparsely covered. Reasons for this bias in the sample distribution include the difficulty of accessing the sparsely populated drylands in the east Kenyan plains, as well as security concerns.

The SDMs are implemented in R (version 4.1.3)[76]. Based on a previous study[77] we used four algorithms frequently and successfully applied in species distribution modelling, i.e. maximum entropy (MaxEnt)[78], support vector machine (SVM)[79], random forest regression (RF)[80] and boosted regression trees (BRT)[81], also known as generalized boosted regression models. We evaluated the models using k-fold cross-validation to tune the different algorithms and focused on the performance parameters area under the curve (AUC[82]) and true skill statistics (TSS[83]). We only considered algorithms for which the majority of species models achieve a TSS value of at least 0.5 and an AUC of at least 0.8 for further analysis. This selection is based on previous studies having either a similar objective or recommending thresholds for these two performance parameters[84,85]. More specifically, we evaluate the models by separating the presence and absence data of each species into a training and a test data set using a k-fold cross-validation, with k = 10. In the final model, all presence and absence data are included. The final model probabilities are calculated for the eight different grass species. The probabilities are calculated over the area given in Fig. S1, with a spatial resolution of 1 km. Only RF and BRT fulfilled the accuracy performance requirements. RF (*randomForest* package)[80] and the BRT model (*dismo* package)[81] are executed with the best parameter combination for each species (Supplementary Table S7). The BRT model uses additionally the function *gbm.step* to define the optimal number of trees and assumes a Bernoulli distribution.

Initially, a set of 37 predictors is used (see Supplementary Table S4). These are correlated with each other using the Spearman spatial correlation, which is a non-parametric test. Pairs or groups of variables with a Spearman's rank of more than 0.6 are reduced to one variable to lower the risk of overfitting and collinearity issues[86,87]. The final model includes the following 11 predictors: annual mean temperature (Bio1), mean diurnal range (Bio2), temperature seasonality (Bio4), annual precipitation (Bio12), precipitation of driest month (Bio14), slope, tree cover (TC), soil texture, cation exchange capacity (CEC), human footprint index (HFI), and distance to waterways (d2ww). Details about the differences between the present and future climate are given in Section "Model verification: future climate response over East Africa under RCP8.5". The true skill statistics (TSS)[83] of the final models range between 0.52 and 0.75 for the RF model (second line in Table 1) and between 0.49 and 0.57 for the BRT model (second line in Supplementary Table S2), which can be considered as a fair to good model performance.

The only species for which the models fail to produce good predictions in the final model is *Pennisetum stramineum*. This poor performance is due to the low number of presence data available for this species. As this issue exceeds the scope of this study, this species is excluded from all further analyses. Since the RF model generally performs slightly better than the BRT model, the results given in this article are based on the RF model. However, the results obtained with the BRT model are provided in the supplementary material. The importance of the different predictors is assessed using two different measures for the RF and the BRT models, respectively. For the RF model, we calculate the "IncNode Purity", which provides the total decrease in node impurities. It indicates how useful a variable is for separating the data into homogeneous groups. A higher IncNode Purity means a more important variable. For the BRT model, we calculate the relative influence of each predictor, which provides an estimate of the predictor's importance in predicting the presence and absence of each grass species in per cent.

In a next step, the 14 models (RF model and BRT model for each of the seven species) are applied to the predictors of the future climate. It is worth noting that the soil- and landscape-related predictors are kept constant; the future climate is reflected exclusively in the bioclimatic predictors. To derive binary suitability maps (suitable vs. unsuitable habitat), a grid point with a probability of presence higher or equal to the TSS of the final model (second line in Table 1 and Supplementary Table S2) is defined as suitable for each grass species, indicating potential presence of that grass species, while grid points with probability values lower than the TSS are considered as potentially unsuitable, indicating potential absence of the species. To understand the changes between the future and the present climate, the following conditions are applied to each grid point: (1) If the models indicate absence of a given species under both present and future conditions, the species is considered to remain absent ("absent"). (2) If the models indicate presence under both present and future conditions, the species is considered to remain present ("no change"). (3) If the models indicate presence under present climate conditions and absence under future conditions, the species is assumed to disappear from this location in the future ("contraction"). Finally, (4) if the models indicate absence under present climate conditions but presence under future conditions, the species is assumed to newly appear at this location in the future ("expansion").

### Model verification: future climate response over East Africa under RCP8.5

To understand future changes in the distribution of East African grasslands, we first examine changes in climate characteristics by presenting the future climate change signal simulated by the RCM. We focus on mean annual temperature and annual accumulated precipitation. As we use only one simulation and one particular global and regional climate model, it is important to compare our climate change signals with other climate modelling results, such as the WorldClim data set (version 1.4)[88]. The main difference between WorldClim and the data set generated for this study consists of the downscaling method. The WRF model performs a dynamical downscaling (see Section "Climate models"), while WorldClim is based on a statistical downscaling of global climate model simulations with horizontal resolutions ranging from 100 to 250 km. The statistical downscaling of WorldClim can be inaccurate in tropical regions, due to the sparsity of observational data[89].

For the comparison, we use the WorldClim data for the years 2061–2080 under RCP8.5, which are closest to the RCP8.5 climate change

scenario for 2071–2100 used in our own climate simulation. Supplementary Figs. S7 and S8 show the difference in mean annual temperature and annual accumulated precipitation between the future and the present-day period for each model included in the WorldClim data set. The top left panel presents the results obtained with our dynamical downscaling approach (CESM-WRF).

Looking at temperature, the ensemble members of WorldClim produce a wide range of different warming patterns over East Africa (Supplementary Fig. S7). One reason for the considerable differences between the model simulations is that the underlying global climate models differ in their climate sensitivity. For example, ACCESS1-0, GFDL-ESM2G, HadGEM2-AO, HadGEM2-CC, HadGEM2-ES, and IPSL-CM5A-LR have a relatively high climate sensitivity and therefore, project a greater temperature increase. In addition, the climate change patterns appear rather smooth, and the effects of geographic features are not visible. This can be expected, as the driving global models are coarsely resolved, and the statistical downscaling approach used in WorldClim relies heavily on observational data, which are sparsely available for our study region. It contrasts with the dynamically downscaled simulation used in this study (Supplementary Fig. S7 top left, CESM-WRF), which shows a more realistic representation of all lakes. For example, the cooling effect of water is visible for Lake Victoria and Lake Turkana. The dynamical downscaling approach is also particularly valuable for properly representing the complex topography of East Africa, with physical consistency between the variables of our model approach, temperature and precipitation, almost maintained. For example, in the region around the Lake Turkana, a comparably small temperature increase is projected for the future compared to the surroundings. This relative cooling is linked to a strong increase in precipitation over the same region (Supplementary Fig. S8 top left, CESM-WRF). It should be noted that the bias correction used in the dynamical downscaling approach of this study could have some implications for physical consistency. Nevertheless, the method is considered to provide robust climate impact scenarios[60,61] and is therefore widely applied in studies on climate change e.g. refs. [62–64] as previously mentioned in Section "Climate models".

Regarding precipitation, most of the WorldClim ensemble members project a general wetting of the study region (black box in Supplementary Fig. S8); exceptions are ACCESS1-0, HadGEM2-AO, GISS-E2-R, MIROC5 and NorESM1-M. In addition, the precipitation patterns differ strongly between different models, in one case even showing inverted signs of the gradient from the coast to the centre of the study region (compare IPSL-CM5A-LR with MPI-ESM-LR). The dynamical downscaling approach (Supplementary Fig. S8 top left, CESM-WRF) likewise predicts a general wetting over the study region, but as with temperature, it shows much more finely resolved precipitation patterns compared to the other simulations. This includes, for example, convective behaviour over the lakes. The precipitation pattern of MRI-CGCM3 model (Supplementary Fig. S8, bottom centre), having the highest horizontal resolution among the global simulations (Supplementary Table S6), is relatively similar to that projected by our CESM-WRF approach.

In summary, the dynamical downscaling results suggest a warming of 2–3.5 °C and a future wetting of the study region under the high-emission scenario RCP8.5. More specifically, pronounced warming is found in north-eastern Kenya and moderate warming in the north-western part. The precipitation pattern also shows a clear structure, with the greatest increase in precipitation projected for the north-western part of Kenya, whereas conditions are expected to become dryer compared to present-day climate in the south-west under RCP8.5. Importantly, the dynamical downscaling results are within the uncertainty range of the WorldClim ensemble members, but with more realistic and finer-scale patterns, almost maintaining the physical consistency between the bioclimatic variables.

### Novel climate in the future under RCP8.5

One important assumption under which machine learning algorithms are applicable is that the distribution and therefore, the range of the training and prediction data set are similar. To check if this assumption is met in our data set, we look for so-called "novel climates" under future climate conditions, which are here defined as climate states outside the range of the present climate within the study area. This is performed by the following two different analyses: The minimum and maximum of all the five bioclimatic variables (see Section "Changes in bioclimatic predictors under the high-emission scenario RCP8.5") under present climate conditions are calculated using (1) only grid points where presence or absence points are available and (2) all grid points in the domain. These numbers are then compared to the future climate (either including only the grid point where presence and absence points are available or for the whole domain). The number of variables that show under future climate conditions higher or lower values than the maximum or minimum value, respectively, in the present climate are summed up at each grid point in the study domain (Supplementary Fig. S9). The range of the future climate variables is mostly covered in the training data set of the past climate for the study region. In the east of the study region, there is a large area where the ranges do not overlap for one variable (mainly mean annual temperature), and in the very northwest and along the coast, two variables exceed the range of the past climate in the future. Since a large part in the east of the study region shows that one variable does not stay within the range of the present climate when considering all grid points in the study area (Supplementary Fig. S9b), only a slight improvement could be obtained with more presence and absence samples in this particular area. Nevertheless, due to the small sample size and the missing overlap of the present and future climate in the eastern part of the study region the results there must be interpreted with caution.

### Limitations of the method

Even though the study is based on a unique and complex model chain that combines global and regional climate models with a species distribution model, it has some limitations which must be kept in mind. In some areas, such as the arid and semi-arid regions in eastern Kenya, we have only sparse coverage of presence and absence data for the different grass species. Accordingly, the presented results and interpretations must be considered with caution. It must further be noted that the climate change signal is obtained from one single realisation of a 30-year climate simulation and one emissions scenario. Obtaining more robust results on the sign of change and particularly of the change in precipitation amounts and patterns would require a larger ensemble of high-resolution climate simulations. Due to the high computational cost of such model simulations, an ensemble simulation is far beyond the possibilities of this study. In any case the comparison of the output of the physical model used here with different WorldClim simulations suggests that the results obtained by our model simulation are within the range of results shown by CMIP5 climate models.

### Data availability

The data to reproduce the study is available at https://doi.org/10.5281/zenodo.8217619. The Global Biodiversity Information Facility (GBIF) data is available at https://www.gbif.org, the sPlot (The Global Vegetation Database) is available at https://www.idiv.de/en/splot.html, the RAINBIO data is available at https://gdauby.github.io/rainbio, the Vegetation Database East Africa (VDEA) is available at http://www.givd.info/ID/AF-00-004, and the Swea-dataveg (A vegetation database for Sub-Saharan Africa) is available at https://kamapu.github.io/posts/sweadataveg/. Bioclimatic variables provided by WorldClim are available at https://www.worldclim.org/data/bioclim.html.

### Code availability

The code to reproduce the study is available at https://doi.org/10.5281/zenodo.8217619.

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

## Acknowledgements

We wish to thank the colleagues and institutions who helped us identify and locate and/or provided us with grass species presence and absence records to develop the database used in this study. They are Dr. Francesco Maria Sabatini (sPlot – The Global Vegetation Database), Dr. Marco Schmidt (East African Plants), and Dr. Miguel Alvarez (SWEA-Dataveg: A vegetation-plot database for Africa). The authors also thank the Swiss National

Supercomputing Centre (CSCS) for providing the computational resources under the project s905 needed to run the regional climate model simulations employed. This research resulted from the Pilot Project Phase of the design for the Wyss Academy for Nature. Generous support by the Wyss Foundation is acknowledged. We also acknowledge financial support from the University of Bern, the Oeschger Centre for Climate Change Research (OCCR), and the Swiss National Science Foundation through projects 200020-200492 and P2BEP2-181837. We also thank the reviewers for their constructive comments, which helped to improve the manuscript.

## Author contributions

The conceptualization was developed by M.M., S.E., A.T.-M.R., M.S., S.J.G.R., and K.H. The global climate model was run by U.B., the regional climate model simulations were run and postprocessed by M.M. and S.J.G.R., the presence and absence data for the different grassland species were gathered by A.T.-M.R., M.S. and A.H., the species distribution modelling was designed and run by M.M. and S.E. The analysis and the interpretation of the results was performed by M.M. with the support of S.E., A.T.-M.R., M.S., S.J.G.R., K.H., U.B., A.H., S.K., and T.F.S. The first version of the manuscript was written by M.M. and S.E., A.T.-M.R., M.S., S.J.G.R., K.H., U.B., A.H., S.K., and T.F.S. took part in the edition and revision of the final version of the manuscript.

## Competing interests

The authors declare no competing interests.
