## [Transparent Peer Review file · Communications Earth & Environment]

Major distribution shifts in key rangeland grasses under a high-emission scenario in East Africa at the end of the 21st century

Corresponding Author: Dr Martina Messmer

Version 0:

Decision Letter:

Dear Dr Messmer,

Your manuscript titled "Reduction in grassland biodiversity under a high-emission scenario in East Africa at the end of the 21st century" has now been seen by 3 reviewers, whose comments are appended below. You will see that they find your work of some potential interest. However, they have raised quite substantial concerns that must be addressed. In light of these comments, we cannot accept the manuscript for publication, but would be interested in considering a revised version that fully addresses these serious concerns.

We hope you will find the reviewers' comments useful as you decide how to proceed. Should additional work allow you to

- address these criticisms (that is, either to incorporate the suggestions or provide a compelling argument why the point made by the reviewer is not valid, or relevant to the editorial threshold as outlined below)

AND

- meet our editorial thresholds as outlined below,

then we would be happy to look at a substantially revised manuscript.

In the following, we list our main editorial concerns.

*Editorial threshold 1: Provide compelling new insights into the grassland biodiversity under future climate change in East Africa

*Editorial threshold 2: Provide a clear and well-structured introduction with sufficient literature review and comprehensive background to justify the necessity of this study

* Editorial threshold 3: Provide in-depth discussion on the implication and underline mechanisms of the findings

If you choose to take up this option, please either highlight all changes in the manuscript text file, or provide a list of the changes to the manuscript with your responses to the reviewers.

If the revision process takes significantly longer than three months, we will be happy to reconsider your paper at a later date, as long as nothing similar has been accepted for publication at Communications Earth & Environment or published elsewhere in the meantime.

Please use the following link to submit your revised manuscript, point-by-point response to the reviewers' comments with a list of your changes to the manuscript text (which should be in a separate document to any cover letter), a tracked-changes version of the manuscript (as a PDF file) and any completed checklist:

Link Redacted

Please do not hesitate to contact us if you have any questions or would like to discuss the required revisions further. Thank you for the opportunity to review your work.

Best regards,

Jinfeng Chang, PhD
Editorial Board Member
Communications Earth & Environment
orcid.org/0000-0003-4463-7778

Alienor Lavergne, PhD
Associate Editor
Communications Earth & Environment

EDITORIAL POLICIES AND FORMAT

If you decide to resubmit your paper, please ensure that your manuscript complies with our editorial policies and complete and upload the checklist below as a Related Manuscript file type with the revised article:

Editorial Policy Policy requirements
(Download the link to your computer as a PDF.)

For your information, you can find some guidance regarding format requirements summarized on the following checklist: (<https://www.nature.com/documents/commsj-phys-style-formatting-checklist-article.pdf>) and formatting guide (<https://www.nature.com/documents/commsj-phys-style-formatting-guide-accept.pdf>).

REVIEWER COMMENTS:

Reviewer #1 (Remarks to the Author):

This paper explores the intriguing subject of modeling the future distribution of seven key grass species in East Africa. While the content is novel, engaging, and readable, there is still room for enhancement in terms of writing and structure, particularly in the introduction and discussion sections. Here are my comments on it and I split them into general comments for each section and specific comments for each line or element.

General comments:

Title:

Your title includes "Reduction in grassland biodiversity", however, you only modeled 7 typical species which cannot reliably represent the grassland biodiversity variation in the future. I recommend you revise the title. Here is an example, "Geographical variation of key grassland species under a high-emission scenario in East Africa at the end of the 21st century".

Introduction:

The literature review in your introduction is not sufficient. Therefore, the demonstration of the research gap and previous research progress in the distribution modeling of grassland species is not clear.

For example, from line 66 to line 69, what's the significance of the distribution and diversity of key grassland species? From line 71 to line 82, SDM is not a new method in biogeography, why did you choose BRT and RF in your research?

Please add more references and restructure your introduction to naturally illustrate your research questions and emphasize your points of innovation.

Methods

1. You did climate modeling in your research. However, why didn't you use the existing future Bioclimatic database, such as CHELSA and WorldClim? What's the improvement of your modeled climatic data compared to the existing database?

2. As for the presence and absence data of your target species, you collected presence data from 5 databases, while only sPlot and VDEA include absence information. Hence, due to this unbalance, I can find an obvious sampling bias between the presence and absence data in Figure S5. In other words, the absence data only distribute in specific regions. Will this introduce bias or errors in your modeling and prediction?

3. A TSS (True Skill Statistics) value larger than 0.4 indicates good model performance. Except for CynDac, all other species have a predictive TSS value lower than 0.5. Have you ever tried to do hyperparameter selection to improve the predictive performance of RF and BRT? I think your model with the best hyperparameter combination will perform better in the prediction of all species.

4. As for the soil- and landscape-related predictors, you mentioned that they were kept constant. I agree that the slope and soil texture will remain constant in the next 50 or 100 years. However, why do you think tree cover, CEC, and HFI will also remain similar in value at the end of the 21st century? Such as tree cover, which is also closely related to tree species distribution, will also be affected by climate change and land use change in the future.

Results and discussion

1. In section 2.2, you described the contribution of different covariates and the spatial distribution of each species. In section 2.3, you discussed about the co-occurrence. I don't think there is more added value to discussing the co-occurrence after the contraction and expansion analysis. Is it better to restructure your Results section and merge sections 2.2 and 2.3 into one section?
2. In section 2.3, you stacked the projected species distribution together and measured the grass biodiversity changes in the future. However, you only included 7 species which is much smaller than the realistic species pool. Hence, I don't think your results can represent any trend related to biodiversity changes in the future.

Conclusion

In the Conclusion section, you discussed a lot about the possible reasons for spatial changes in grass species. You included many extended analyses of the results and their comparison to the literature. However, I think it reads more like a Discussion.

Specific comments:

1. In line 47, you mentioned that "climate change, land conversion, rising CO₂ levels, and fire disturbance are major drivers of grassland change". But why did you only consider climate change and rising CO₂ levels in your research?
2. In Figure 1b, I cannot find the positive value (red) in your plot, please check if the legend is the correct one.
3. In Table 1, I think it is better to add one column to demonstrate the overall (mean) variable importance for all covariates. Because you described the importance of each covariate not only at the species level but also across all species.
4. In lines 172 to 176, you got the results about the areas projected to lose *T. triandra* may experience a decline in grassland ecosystem functions in the future. However, I don't think this is enough evidence for this conclusion according to your research. Because you didn't state if this is the only grass species that contributes to pasture resilience. If the area experiencing a range contraction of this species will not be colonized by other species that could prove a similar ecosystem functioning. In summary, you need more arguments and more references here.
5. In Figure 3 and Figure S3, you plotted the current species richness and the future-present. However, same as the spatial pattern of current co-occurrence, future co-occurrence is also necessary for your analysis. Hence, please also add the subplot of the future co-occurrence.
6. In lines 238 to 240, you concluded why various grass species would decline due to human disturbance and land degradation. However, in your research, landscape- and soil-relevant variables were kept constant for the projection. How did you get this conclusion?
7. In line 308, how did you divide the study region into physiographic units? Is there any reference or existing map?
8. In line 324, there might be some information missing.
9. From line 325 to line 335, instead of 18, I only found 17 environmental variables.
10. In line 398, please add the information on the R version you used.
11. I didn't find the relevant text about Table S4 in your manuscript.
12. In line 408, there might be some information missing.
13. In lines 409 to 411, you mentioned your variable selection based on Spearman correlation. Why didn't you show the results of the Spearman correlation?
14. In lines 417 to 420, you talked about the model performance of RF and BRT. Will it be better to move this part to the Results section?
15. In Figure S7 and Figure S8, you used the word "difference" to name your figures. However, you didn't calculate the difference between GCMs, yet show the projected results of temperature or precipitation. I think it is better to revise the names of these two figures.
16. In line 456, you explained dynamical downscaling (see Sect. 6.3), but I didn't find a sufficient explanation about dynamical downscaling in section 6.3. Please add more information and make it more readable and easier for readers to find the dynamic downscaling part which is quite important in your research.
17. In Section 6.6, you interpret the problem of extrapolating. I think it is better to move part of it to 2.3 as a discussion.
18. In Figure S9, your color bar ranges from 0 to 5, but I didn't find any regions that were more than 2. Is it better to remake your legend?

Reviewer #2 (Remarks to the Author):

The manuscript "Reduction in grassland biodiversity under a high-emission scenario in East Africa at the end of the 21st century" integrates global and regional climate models with a machine learning-based species distribution model. This approach is used to assess the impact of climate change on East African grassland biodiversity, indicating a potential decrease in species co-occurrence and adverse effects on pastoralism and wildlife under a high-emission scenario by the end of the century.

While I am not an expert in modeling, I believe that this paper has the potential to significantly contribute to our understanding of the impacts of climate change on grassland diversity in East Africa. However, the current structure of the paper detracts from its readability and clarity, making it challenging to comprehend. Additionally, I find that the paper lacks

sufficient information and discussion on the ecological implications of the findings. I recommend a thorough revision to address these issues. The paper should be considered for publication only after such a revision.

1. Introduction

The structure of the introduction currently lacks clarity, and I believe it could greatly benefit from a more defined outline and additional content. To improve this, I suggest the following structure:

1. First Paragraph: Retain the existing content, setting the stage for the topic.
2. Second Paragraph: Expand on the grasses being studied - their ecological significance, the ecosystems they inhabit, and factors influencing their growth, including soil properties. This paragraph could also introduce the variables used in the models, establishing a foundation for the subsequent discussion.
3. Third Paragraph: Address future climate changes and their anticipated impacts on grassland ecosystems and the specific grasses under study. Reference existing studies and statistically downscaled global model outputs, and detail the predictions these models make for the region in question.
4. Fourth Paragraph: The one with the current opening line, "Here...", its informative and clear, I like it.
5. Hypotheses and Expectations: It would be beneficial to include what the authors hypothesized or expected to find. This could be integrated into the fourth paragraph or presented as a separate one, providing readers with a clearer understanding of the study's direction.
6. Last Paragraph: The current final paragraph feels abruptly short and somewhat disconnected. Why is only one of the models mentioned? Distributing some of this content to the preceding paragraph might also help in maintaining balance and coherence.

Overall, by reorganizing the introduction in this manner, it would not only enhance the flow and clarity but also provide a comprehensive background and set a clear direction for the study.

Specific comments:

Line 41: Maybe rephrase, for example like this, to make it clearer: "Grasslands are a crucial ecosystem in East Africa, serving as habitats and grazing grounds for wildlife, which supports tourism, and providing forage for livestock".

Line 56 and 60: Is it required from the journal? Otherwise I would remove it: "(Method)". Its clear that this part can be found in the methods.

Line 56-58: This sentence belongs to the end of the paragraph/section. Here I would focus on what the authors actually did.

Line 72-75: This is important, maybe you can elaborate a bit more here (or in the new paragraph on the grasses and their ecology).

Line 76 and whole paragraph: Okay, but what about the other models you used?

2. Results and Conclusion

I find myself somewhat perplexed with the structure of this manuscript. The current format merges the results and discussion into one section, despite it being labeled solely as 'Results'. At the same time, there is a "Conclusion" section that discusses a bit more. For enhanced clarity and comprehensibility, I propose a restructuring of these sections. In general, the Results should present the data and findings clearly and concisely, without interpretation. The Discussion, on the other hand, interprets these findings, discussing their implications, limitations, and how they fit into the broader context of the field. Therefore, I propose the authors to write a distinct and detailed 'Results' section, focusing exclusively on presenting the findings. This should be followed by a separate 'Discussion' section, where the findings are interpreted in-depth. In this section, longer paragraphs should be dedicated to elaborating on the meanings of the findings, their broader implications, and how they contribute to the field. Such a separation would provide a clearer path for readers to navigate the paper. This would enhance the overall readability and impact of the paper.

Line 281: Figure 1 and 3, the descriptions at the top are difficult to read. It would be beneficial to increase the font size for this figure overall.

Reviewer #3 (Remarks to the Author):

The authors present an interesting study combining species distribution models and machine learning to predict the potential distributions of seven key grass species in East Africa under future climate change scenarios. However, several critical aspects require attention. First, the research questions or knowledge gaps addressed by the study are not clearly articulated. The study appears more like a data analysis exercise than a targeted investigation aimed at addressing specific, novel research questions. The authors need to clarify what new insights their study provides and which specific knowledge gaps it aims to fill. Secondly, the manuscript's structure requires improvement, particularly in the results, discussion, and conclusions sections. Sometimes, authors integrate results with discussion for better logical flow, but in this version, it doesn't seem to aid understanding. Additionally, the manuscript lacks a thorough and insightful discussion. A significant portion of the conclusions reads more like a discussion, which needs to be addressed for clearer demarcation and effectiveness.

Abstract:

The abstract should articulate the knowledge gaps and novelty more clearly. It currently lacks specific questions. What are the key questions being addressed? The results parts need clearer articulation, particularly regarding the 'reduction of grassland biodiversity.' Terms like 'large parts,' 'high-emission,' and 'end of the 21st century' are too vague. How could authors compare these words? With the current or low-emission climate scenarios? In the last sentence, how are the study's

insights particularly relevant to pastoralism and wildlife in East Africa? The transitional sentences are missed.

Introduction:

Lines 41-53: The introduction lacks specific research questions. Why is East Africa's grassland ecosystem unique in the context of climate change? What specific knowledge gaps does this study aim to fill?

Lines 76-82: The transition to discussing SDMs is not smooth. It needs a bit of work.

Results:

Lines 118-121: Tree cover appears to be an important covariate for predicting grass species distribution. How does tree cover generally respond to a high-emission scenario like RCP8.5?

Lines 121-122: It's unclear whether the loss of grass species is due to increased tree cover under RCP8.5 or directly due to RCP8.5.

Line 123: Please provide full names or descriptions for BIO1, BIO2, and BIO4 to aid reader comprehension.

Conclusion:

Lines 235-248: This section reads more like a discussion. Also, consider including how livestock grazing intensity and land use intensification in East Africa might interact with RCP8.5.

Overall, I feel that the title might be more accurately phrased as 'Grassy Species' rather than 'Grassland Diversity.'

Methods:

Lines 526-552: The acknowledgment of study limitations is appreciated. Integrating this with the discussion section to outline future research directions could be more effective.

Tables:

Table 1: Percentages exceeding 100% for the most important predictor are confusing. Standardizing these percentages for comparability among species would be clearer.

Figures:

Figure 3: Consistency with previous figures, including labels like A/B for each panel, would be helpful. Also, please specify the units on each map.

Communications Earth & Environment is committed to improving transparency in authorship. As part of our efforts in this direction, we are now requesting that all authors identified as 'corresponding author' create and link their Open Researcher and Contributor Identifier (ORCID) with their account on the Manuscript Tracking System prior to acceptance. ORCID helps the scientific community achieve unambiguous attribution of all scholarly contributions. You can create and link your ORCID from the home page of the Manuscript Tracking System by clicking on 'Modify my Springer Nature account' and following the instructions in the link below. Please also inform all co-authors that they can add their ORCID to their accounts and that they must do so prior to acceptance.

Author Rebuttal letter: The author's response to these comments can be found at the end of this file.

Version 1:

Decision Letter:

Dear Dr Messmer,

Your manuscript titled "Major distribution shifts in key rangeland grasses under a high-emission scenario in East Africa at

the end of the 21st century" has now been seen by our reviewers, whose comments appear below. In light of their advice we are delighted to say that we are happy, in principle, to publish a suitably revised version in Communications Earth & Environment.

We therefore invite you to revise your paper one last time to address the remaining concerns of our reviewers. At the same time we ask that you edit your manuscript to comply with our format requirements and to maximise the accessibility and therefore the impact of your work.

EDITORIAL REQUESTS:

****Please take care to match our formatting and policy requirements. We will check revised manuscript and return manuscripts that do not comply. Such requests will lead to delays. ****

SUBMISSION INFORMATION:

OPEN ACCESS:

Communications Earth & Environment is a fully open access journal. Articles are made freely accessible on publication. For further information about article processing charges, open access funding, and advice and support from Nature Research, please visit <https://www.nature.com/commsenv/open-access>

Link Redacted

Best regards,

Martina Grecequet, PhD
Associate Editor,
Communications Earth & Environment
@CommsEarth

Jinfeng Chang, PhD
Editorial Board Member
Communications Earth & Environment

REVIEWERS' COMMENTS:

Reviewer #1 (Remarks to the Author):

Thanks for the careful revision from all the authors. After reviewing the responses, I am pleased to see that most of my comments have been addressed thoroughly. However, there are still two main areas that require further attention.

Regarding the abstract, one more sentence about the research gap or the significance of your research following the first sentence is necessary for the flow and logic of your abstract. Not only how important the grassland habitat is, but also you need to demonstrate how important is modeling the distribution of key grassland species under climate change.

Regarding the introduction, while I appreciate the efforts made to revise the introduction, I believe there is room for improvement in its structure and flow.

Specifically, the second paragraph illustrated the significance of the 7 key species in your study. It may come from the comments of other reviewers. However, I think you should generally demonstrate how important the key grassland species are to the whole grassland ecosystem, rather than focus solely on these 7 species. Some of the detailed explanations about the seven species could be moved to the methodology section or an appendix.

Besides, the linking and flow of the introduction is not natural enough. You used a lot of stilted sentences and phrases, such as "The current state of research reveals (line 104)", "The combination of these factors highlights a research gap that (line 112)", "To address these data and knowledge gaps (line 115)", etc. These phrases disrupt the natural flow of the text. Consider rephrasing these sentences to create a more engaging and fluid narrative. Highlighting the research gap is important, but it should be done in a way that seamlessly draws the reader into the study.

Finally, you spent too many lines illustrating your research objectives and contents (from line 115 to line 151) which, in my opinion, should be shorter. Even though, your research questions are still not clear. Hence, please clarify your research questions before illustrating your research objectives. From my side, the last paragraph of the introduction (from line 152 to line 158) is also redundant and unnecessary.

Regarding your methodology, you explained why you modeled climatic data and the improvement of it compared to the existing databases. However, I still doubt the improvement of using high temporal resolution simulated data. The justification for using high temporal resolution simulated data remains unclear. Since the distribution and growth of grassland species are influenced by climatic conditions, rather than hourly weather conditions. A species will not present or disappear due to hourly, daily, or even monthly changing precipitation or temperature. Hence, in my opinion, you should at least elaborate on the benefits of using fine temporal predictors in SDMs or cite more relevant references to support your methodology.

Once these issues are addressed, I believe that the manuscript will be ready for acceptance. Thank you for your continued efforts in improving the quality of the paper.

Reviewer #2 (Remarks to the Author):

I am pleased to report that I am satisfied with the implemented changes. The revisions have effectively addressed all the concerns and suggestions previously highlighted. The improvements enhance the clarity, coherence, and overall quality of the work.

Reviewer #3 (Remarks to the Author):

Reading through the MS, the study has substantially improved. I only have a few comments.

Lines 127-133, it looks like unnecessary text here.

In the results, I would like to remove the quoted labels, such as "(Bio1)", "(A1)", "(B1)", and so on. The conclusions are a bit farfetched and redundant, they can be condensed more.

Author Rebuttal letter: The author's response to these comments can be found at the end of this file.

Replies to Reviewers

Reviewer #1:

Introduction:

The literature review in your introduction is not sufficient. Therefore, the demonstration of the research gap and previous research progress in the distribution modeling of grassland species is not clear.

We worked extensively on the introduction and restructured it significantly. In particular, we have focused on the motivation of our study and better embedded our research question. We have also added a paragraph on the importance of the grassland species studied.

From line 71 to line 82, SDM is not a new method in biogeography, why did you choose BRT and RF in your research?

We thank the reviewer for this question. Based on an extensive SDM literature review by Eckert et al. (2020) we used four different machine learning algorithms. A combination of more novel machine learning algorithms and more traditional statistical approaches was chosen based on obtained accuracies. The initial four algorithms were: support vector machine (SVM), random forest regression (RF), boosted regression trees (BRT), and maximum entropy (MaxEnt). Based on an accuracy threshold requirement of TSS>0.5 we then selected those algorithms that fulfilled this requirement. Only RF and BRT fulfilled this requirement.

We have added the missing information on the model selection in the methods section (Section 7.4, line 522 - line 538).

Please add more references and restructure your introduction to naturally illustrate your research questions and emphasize your points of innovation.

As stated above, we have substantially restructured the introduction, and hope that the reviewers find the current version of the manuscript more logical and easier to follow.

Methods:

1. You did climate modeling in your research. However, why didn't you use the existing future Bioclimatic database, such as CHELSA and WorldClim? What's the improvement of your modeled climatic data compared to the existing database?

Existing databases, such as WorldClim and CHELSA, are based on global climate model simulations with a spatial resolution of 100 km or more. These data are statistically downscaled based on station data or reanalysis output to provide more detailed and regional information. However, these data sets are not able to capture regional climate related to, for example, topography or lake effects. To obtain more realistic regional climate projections, dynamical downscaling is required. We have done this using the WRF regional climate model. Dynamical downscaling yields physically consistent variables, as temperature and precipitation variables are interdependent. However, this dependence is broken when statistical corrections are applied to the two variables separately. In addition, regional climate models have a fine temporal resolution, in our case, an hourly output. This high temporal resolution allows much more precise values to be obtained for the diurnal range, for example, as it is based on an actual day rather than a minimum and maximum temperature of monthly values. A detailed description of the added value of the regional climate model output can be found in the Methods section (Section 7.3), Figures S7 and S8.

2. As for the presence and absence data of your target species, you collected presence data from 5 databases, while only sPlot and VDEA include absence information. Hence, due to this unbalance, I can find an obvious sampling bias between the presence and absence data in Figure S5. In other words, the absence data only distribute in specific regions. Will this introduce bias or errors in your modeling and prediction?

It is well known that working with presence-absence data leads to more accurate species distribution models. This is particularly the case for true absence data collected in a systematic way. Absence data improve model accuracy by providing a clearer distinction between suitable and unsuitable areas.

However, such data are scarce, often only available for small areas. An option is to work with pseudo-absence data, but they have the drawback of potentially coinciding with locations where the species actually occur. Thus, we decided to work with real absences after checking their availability for this analysis. The absence data were systematically collected in so-called vegetation relevés for different altitudes (along slopes of different topographic features). As a result, the elevation space, which is an important factor in habitat suitability, is well sampled. In addition, we have added a new figure S6 in the Supplement, which shows the species as columns and the different datasets as rows. This figure shows the selected presence and absence points for each species and dataset.

Overall, the number of presence and absence data is relatively balanced, with 42% of presence data and 58% of absence data. We do agree that for three species the distribution is rather unbalanced, i.e. *Digitaria milanjana*, *Cenchrus megianus*, and *Pennisetum stramineum* (see Table S1). *D. milanjana* and *C. megianus* were modelled with acceptable TSS in all machine learning algorithms. Both are common grass species in Kenya and not rare. Thus, we decided to keep these species in the study and use all data points available. However, *P. stramineum* had a very small, unbalanced number of presence and absence points, and we decided to exclude it from the study. We have included this information in Section 7.4.

3. Have you ever tried to do hyperparameter selection to improve the predictive performance of RF and BRT? I think your model with the best hyperparameter combination will perform better in the prediction of all species.

We tuned the RF using values of 'mtry' of 1, 2 and 3, 'ntree' of 100 to 2500 and 'nodesize' of 1 to 6. This resulted in slightly different optimal values for the three variables depending on the species when optimising the AUC. For each species, we used the optimal setting, which can be found in Table S7 of the Supplement. For the BRT model, we changed tree.complexity from 5 to 30, learning.rate between 0.001, 0.005 and 0.01, and bag.fraction from 0.5 to 0.75. We optimised for AUC in the test dataset and used a different optimal setting for each species. These can also be found in Table S7 in the Supplement.

We modified the method section accordingly (Section 7.4, line 522 - line 538).

4. As for the soil- and landscape-related predictors, you mentioned that they were kept constant. I agree that the slope and soil texture will remain constant in the next 50 or 100 years. However, why do you think tree cover, CEC, and HFI will also remain similar in value at the end of the 21st century? Such as tree cover, which is also closely related to tree species distribution, will also be affected by climate change and land use change in the future.

We thank the reviewer for this interesting thought. It is true that land use and land cover will change in the future. However, these changes are strongly dependent on human decisions and not on physical parameters, and therefore, predicting these changes is much more complex and diverse than predicting climate change. To understand how climate and its changes might affect the distribution of different grass species, we have changed only these fields and held all others constant. This allows us to relate changes in grass species to changes due to climate change, which is the main research question of our study. We made this clearer with the new structure of our introduction and discussion. In the latter, we also discuss that climate change is not the only factor affecting grassland ecosystems. It is beyond the scope of this publication to include all the different factors that may affect grassland species. This publication provides a baseline and can be expanded in complexity in follow-up studies. We have moved some text from the section "Limitations" to the conclusion and outlook section of the new version of the manuscript.

Results and discussion

1. In section 2.2, you described the contribution of different covariates and the spatial distribution of each species. In section 2.3, you discussed about the co-occurrence. I don't think there is more added value to discussing the co-occurrence after the contraction and expansion analysis. Is it better to restructure your Results section and merge sections 2.2 and 2.3 into one section?

Thank you for this suggestion, we have merged the two sections, and the co-occurrence now provides the summary of "Patterns and trends of seven key grass species". See the new Section 2.2.

2. In section 2.3, you stacked the projected species distribution together and measured the grass biodiversity changes in the future. However, you only included 7 species which is much smaller than the realistic species pool. Hence, I don't think your results can represent any trend related to biodiversity changes in the future.

Thank you for this comment. We propose to use these 7 species as a first order indicator of biodiversity change, as the species surveyed are widespread and important savannah grasses, and could therefore serve as a first indication of East African savannah biodiversity. We appreciate that this needs to be adjusted, particularly as the section heading does not represent the first-order estimate. As we have acted on your suggestion to merge the two sections 2.2 and 2.3, this issue is hopefully resolved.

Conclusion

In the Conclusion section, you discussed a lot about the possible reasons for spatial changes in grass species. You included many extended analyses of the results and their comparison to the literature. However, I think it reads more like a Discussion.

We have split the results part into a pure results part and a new discussion section, where we have used most of our old conclusion section. We hope that with the new structure of the manuscript it is easier to follow our storyline.

Specific comments:

1. In line 47, you mentioned that "climate change, land conversion, rising CO₂ levels, and fire disturbance are major drivers of grassland change". But why did you only consider climate change and rising CO₂ levels in your research?

Thank you for this comment, as described in one of the main points above, we want to understand the impact of direct climate change on the different grassland species. This is why we are holding all other variables constant, even though some of them are likely to change in the future.

2. In Figure 1b, I cannot find the positive value (red) in your plot, please check if the legend is the correct one.

Thank you for pointing this out, we have adjusted the colour bar to include only the colours shown in the figure.

3. In Table 1, I think it is better to add one column to demonstrate the overall (mean) variable importance for all covariates. Because you described the importance of each covariate not only at the species level but also across all species.

This is a good suggestion, we have added an extra column for the overall variable importance.

4. In lines 172 to 176, you got the results about the areas projected to lose *T. triandra* may experience a decline in grassland ecosystem functions in the future. However, I don't think this is enough evidence for this conclusion according to your research. Because you didn't state if this is the only grass species that contributes to pasture resilience. If the area experiencing a range contraction of this species will not be colonized by other species that could prove a similar ecosystem functioning. In summary, you need **more arguments and more references** here.

We have changed the wording of 'ecosystem functions' to 'ecosystem services to pastoralists' as we are specifically referring to the quality of grazing if *T. triandra* disappears.

5. In Figure 3 and Figure S3, you plotted the current species richness and the future-present. However, same as the spatial pattern of current co-occurrence, future co-occurrence is also necessary for your analysis. Hence, please also add the subplot of the future co-occurrence.

Thank you for this comment, we have added the future co-occurrence to the two figures, so that we have three panels now in the new Figures 3 and S3

6. In lines 238 to 240, you concluded why various grass species would decline due to human disturbance and land degradation. However, in your research, landscape- and soil-relevant variables were kept constant for the projection. How did you get this conclusion?

We added this sentence to the discussion to point out that in addition to climate change, which we are analysing, other anthropogenic factors may be affecting the grass species. This statement is supported by other studies cited. To make this clearer, we have rephrased this statement.

7. In line 308, how did you divide the study region into physiographic units? Is there any reference or existing map?

We realised that the link we have added to section 7.1, where we describe the rationale of the physiographic units, does no longer work. We have adjusted the link accordingly to:
<https://infonet-biovision.org/agro-ecological-zones/aezs-kenya-system>

There you can find the seven different agro-ecological zones, as shown in the figure below:

The agro-ecological zones (AEZ) refer to “dividing an area of land into smaller units with similar characteristics related to land suitability, potential production, and environmental impact. An AEZ is a land resource mapping unit defined in terms of climate, landform, soils, and/or land cover, having specific potentials and constraints for land use (FAO, 1996). The key elements for defining an AEZ are the growing period, temperature regime, and soil mapping unit.”
Source: <https://infonet-biovision.org/agro-ecological-zones/aezs-kenya-system>, last access on April, 25th 2024.

We have added the explanation of agro-ecological zones to the paper as: “Agro-ecological zones categorize a larger land area into smaller sections that share similar traits regarding land suitability, potential production, and environmental impact.”

8. In line 324, there might be some information missing.

Thank you for pointing this out. The reference to Table S4 in the Supplement was lost and we have added it again.

9. From line 325 to line 335, instead of 18, I only found 17 environmental variables.

This is correct, we missed even two variables in the text. This has been corrected now.

10. In line 398, please add the information on the R version you used.

We have added the R version in the new manuscript.

11. I didn't find the relevant text about Table S4 in your manuscript.

For some reason the link to Table S4 was always lost in the uploaded files. We have now added it to the manuscript and you should be able to find the link between Table S4 and the test in the manuscript.

12. In line 408, there might be some information missing.

Thank you for pointing this out, again the reference to Table S4 was lost and is now added again.

13. In lines 409 to 411, you mentioned your variable selection based on Spearman correlation. Why didn't you show the results of the Spearman correlation?

We have compiled a cross table with 37 x 37 variables and highlighted the Spearman correlation. If variables had a correlation higher/lower than ± 0.6 , a representative variable was selected. Using this approach, we reduced the number of variables from 37 to 11 to reduce the risk of overfitting the SDM. As this table is very large and contains a lot of detail, we decided to focus on more important figures and tables in the supplement and therefore did not include it in the manuscript.

14. In lines 417 to 420, you talked about the model performance of RF and BRT. Will it be better to move this part to the Results section?

We have rewritten the method part with respect to the species distribution modelling and we have removed this part of the text, as it is no longer applicable.

15. In Figure S7 and Figure S8, you used the word “difference” to name your figures. However, you didn’t calculate the difference between GCMs, yet show the projected results of temperature or precipitation. I think it is better to revise the names of these two figures.

Thank you for your thoughtful comment regarding the naming of Figures S7 and S8. In these figures, the term "difference" is used to denote the contrast between projected future climate conditions and the present climate, rather than differences among various GCMs. We believe this terminology accurately reflects the content displayed in the figures. Therefore, we prefer to retain the current naming, as it directly aligns with the analyses presented.

16. In line 456, you explained dynamical downscaling (see Sect. 6.3), but I didn’t find a sufficient explanation about dynamical downscaling in section 6.3. Please add more information and make it more readable and easier for readers to find the dynamic downscaling part which is quite important in your research.

We have added the following sentence to Section 7.3 and hope that this helps to understand what we mean with dynamical downscaling:

“The regional model solves the fundamental physical equations governing the atmospheric dynamics, e.g., thermodynamics, radiation and moisture transport, on this grid and thus obtains a dynamical downscaling.”

17. In Section 6.6, you interpret the problem of extrapolating. I think it is better to move part of it to 2.3 as a discussion.

Thank you for this consideration. Nevertheless, we view the novel climates primarily as an issue of the methods and therefore, we have decided to keep it in this part of the manuscript.

18. In Figure S9, your color bar ranges from 0 to 5, but I didn’t find any regions that were more than 2. Is it better to remake your legend?

We use the number 5 in the color bar as this is in fact the maximum theoretical number that you can obtain, since we use 5 climate related variables that we are changing for the future. We have added a sentence in Section 7.6 to point out that maximal two variables exceed the range of the present climate in the future.

References:

Eckert, S., Hamad, A., Kilawe, C. J., Linders, T. E. W., Ng, W.-T., Mbaabu, P. R., Shiferaw, H., Witt, A., and Schaffner, U.: Niche change analysis as a tool to inform management of two invasive species in Eastern Africa, *Ecosphere*, 11, e02987, <https://doi.org/10.1002/ecs2.2987>, 2020.

Reviewer #2:

While I am not an expert in modeling, I believe that this paper has the potential to significantly contribute to our understanding of the impacts of climate change on grassland diversity in East Africa. However, **the current structure of the paper detracts from its readability and clarity, making it challenging to comprehend. Additionally, I find that the paper lacks sufficient information and discussion on the ecological implications of the findings.** I recommend a thorough revision to address these issues. The paper should be considered for publication only after such a revision.

1. Introduction

The structure of the introduction currently lacks clarity, and I believe it could greatly benefit from a more defined outline and additional content. To improve this, I suggest the following structure:

1. First Paragraph: Retain the existing content, setting the stage for the topic.
2. Second Paragraph: Expand on the grasses being studied - their ecological significance, the ecosystems they inhabit, and factors influencing their growth, including soil properties. This paragraph could also introduce the variables used in the models, establishing a foundation for the subsequent discussion.
3. Third Paragraph: Address future climate changes and their anticipated impacts on grassland ecosystems and the specific grasses under study. Reference existing studies and statistically downscaled global model outputs, and detail the predictions these models make for the region in question.
4. Fourth Paragraph: The one with the current opening line, "Here...", its informative and clear, I like it.
5. Hypotheses and Expectations: It would be beneficial to include what the authors hypothesized or expected to find. This could be integrated into the fourth paragraph or presented as a separate one, providing readers with a clearer understanding of the study's direction.
6. Last Paragraph: The current final paragraph feels abruptly short and somewhat disconnected. Why is only one of the models mentioned? Distributing some of this content to the preceding paragraph might also help in maintaining balance and coherence.

We would like to thank the reviewer for the effort to improve the structure of our paper. This is much appreciated and has helped to improve the structure, flow and clarity of the introduction. We have worked hard on the introduction to follow this structure. We hope that it is now more fluent and that the aim of our study has become clear.

Specific comments:

Line 41: Maybe rephrase, for example like this, to make it clearer: "Grasslands are a crucial ecosystem in East Africa, serving as habitats and grazing grounds for wildlife, which supports tourism, and providing forage for livestock".

Thank you for your suggestion. As we have completely rewritten the introduction, your suggestion is no longer applicable, but we hope that the current version is more fluent and logical.

Line 56 and 60: Is it required from the journal? Otherwise I would remove it: "(Method)". Its clear that this part can be found in the methods.

We agree and have removed the reference to the method section in the introduction.

Line 56-58: This sentence belongs to the end of the paragraph/section. Here I would focus on what the authors actually did.

During the rewriting process, we have removed this sentence.

Line 72-75: This is important, maybe you can elaborate a bit more here (or in the new paragraph on the grasses and their ecology).

We added a lot more text about the grass species and their importance for wildlife and pastoralists in the 2nd paragraph of the introduction, which should provide more context.

Line 76 and whole paragraph: Okay, but what about the other models you used?

Thank you for this comment. We have removed it from the introduction as we feel it does not really belong there.

2. Results and Conclusion

I find myself somewhat perplexed with the **structure of this manuscript**. The current format merges the results and discussion into one section, despite it being labeled solely as 'Results'. At the same time, there is a "Conclusion" section that discusses a bit more. For enhanced clarity and comprehensibility, I propose a restructuring of these sections. In general, the Results should present the data and findings clearly and concisely, without interpretation. The Discussion, on the other hand, interprets these findings, discussing their implications, limitations, and how they fit into the broader context of the field. Therefore, I propose the authors to write a distinct and detailed 'Results' section, focusing exclusively on presenting the findings. This should be followed by a separate 'Discussion' section, where the findings are interpreted in-depth. In this section, longer paragraphs should be dedicated to elaborating on the meanings of the findings, their broader implications, and how they contribute to the field. Such a separation would provide a clearer path for readers to navigate the paper. This would enhance the overall readability and impact of the paper.

We have separated the results from the discussion and moved most of the conclusions to the discussion. We hope that this substantial reorganisation of the manuscript will improve readability and better highlight the aim of our study.

Line 281: **Figure 1 and 3**, the descriptions at the top are difficult to read. It would be beneficial to increase the font size for this figure overall.

Thank you for pointing this out. We have adjusted the font size of the figures to increase the readability as suggested.

Reviewer #3:

The authors present an interesting study combining species distribution models and machine learning to predict the potential distributions of seven key grass species in East Africa under future climate change scenarios. However, several critical aspects require attention. First, the **research questions or knowledge gaps addressed by the study are not clearly articulated**. The study appears more like a data analysis exercise than a targeted investigation aimed at addressing specific, novel research questions. The authors need to clarify what new insights their study provides and which specific knowledge gaps it aims to fill. Secondly, the manuscript's **structure requires improvement**, particularly in the **results, discussion, and conclusions** sections. Sometimes, authors integrate results with discussion for better logical flow, but in this version, it doesn't seem to aid understanding. Additionally, the manuscript **lacks a thorough and insightful discussion**. A significant portion of the conclusions reads more like a discussion, which needs to be addressed for clearer demarcation and effectiveness.

Abstract:

The abstract should articulate the knowledge gaps and novelty more clearly. It currently lacks specific questions. What are the key questions being addressed? The results parts need clearer articulation, particularly regarding the 'reduction of grassland biodiversity.' **Terms** like 'large parts,' 'high-emission,' and 'end of the 21st century' are **too vague**. How could authors compare these words? With the current or low-emission climate scenarios? In the last sentence, how are the study's insights particularly relevant to pastoralism and wildlife in East Africa? The transitional sentences are missed.

We have rewritten parts of the abstract to make the key question clearer, but given the requirement for 150 words, we cannot elaborate much more on this. We further have removed the word 'high-emission' as the RCP8.5 scenario implies that we are using a high-emission scenario and therefore, there is no need to specify this twice in the abstract. Regarding your concern about the end of the 21st century, we have added "for the last 30 years of the 21st century". In climate research, averaging over such a period is standard to capture climatic trends effectively. We feel that further specification beyond this would not enhance the accuracy due to the nature of climate variability over time. We hope that the abstract is now clearer for the reader.

Lines 41-53: The **introduction lacks specific research questions**. Why is East Africa's grassland ecosystem unique in the context of climate change? What specific knowledge gaps does this study aim to fill?

We have worked a lot on the introduction and hope that the new version of the manuscript makes the research question and our motivation clear.

Lines 76-82: The transition to discussing SDMs is not smooth. It needs a bit of work.

We agree and have removed this part as it is more method-related and does not necessarily need to be in the introduction and to reduce the length of the introduction.

Results:

Lines 118-121: Tree cover appears to be an important covariate for predicting grass species distribution. How does tree cover generally respond to a high-emission scenario like RCP8.5?

That is an interesting question. To answer it, we would need to model vegetation change in response to climate change. This would require the coupling of a global dynamic vegetation model, which is beyond the scope of this study. In rewriting our introduction, we have made our research question more specific, and we hope it is now clear that we are focusing on the direct influence of climate change on the different key grass species. This means that we ignore indirect changes in tree cover, changes in land use, but also changes in soil properties. We have also added a small part in the outlook section, that such an analysis might be an interesting follow-up study.

Lines 121-122: It's unclear whether the loss of grass species is due to increased tree cover under RCP8.5 or directly due to RCP8.5.

As we do not change tree cover, it can be assumed that the loss of grass species is related to the climatic changes due to RCP8.5. There is a particularly strong reduction in those regions where there is an increase in precipitation (i.e. Lake Turkana region). The model suggests a decrease in grass species because, at similar climatic values, there is no data point of presence, either because there is

too much rain for the grass species to grow in the first place, or because there is no grass species present because the trees have an advantage in these climatic conditions and therefore grasses cannot occupy this habitat. With our model approach it is not possible to decide which of the two options it is, due to the complex and non-linear relationship between the variables.

Line 123: Please provide full names or descriptions for BIO1, BIO2, and BIO4 to aid reader comprehension.

The important bioclimatic predictors (Bio1, Bio2, Bio4, Bio12, Bio14) are introduced in Section 2.1. Additionally, there is a supplementary table (Table S4) that lists the abbreviations. We believe writing out the indicators will decrease readability. Thus, we prefer working with the abbreviations in the main paper and providing a supplementary with the full names of the bioclimatic predictors.

Conclusion:

Lines 235-248: This section reads more like a discussion. Also, consider including how livestock grazing intensity and land use intensification in East Africa might interact with RCP8.5.

Yes, it is true that the conclusions were more like a discussion. Therefore, we restructured the previous Results and Conclusions into Results, Discussion, and Conclusion sections in the new version. We acknowledge that there are significant uncertainties in projecting changes in livestock grazing intensity and land use intensification, as they depend on human decision-making, which may indeed be unpredictable. However, given the scope and objectives of our current study, including such projections would go beyond our central aims and feasibility constraints. We have therefore decided not to include this aspect in our discussion.

Overall, I feel that the title might be more accurately phrased as 'Grassy Species' rather than 'Grassland Diversity.'

We agree that the 'grassland diversity' is not an accurate phrasing, so we changed our title to: "Major distribution shifts in key rangeland grasses under a high-emission scenario in East Africa at the end of the 21st century".

Methods:

Lines 526-552: The acknowledgment of **study limitations is appreciated. Integrating this with the discussion** section to outline future research directions could be **more effective**.

Thank you for this comment, we have decided to move some of the limitations to the Conclusions and Outlook section, as it is now written as a further development of our study in a potential follow-up publication.

Tables:

Standardizing these percentages for comparability among species would be clearer.

Thank you for this comment. We have implemented this as suggested.

Figures:

Figure 3: Consistency with previous figures, including labels like A/B for each panel, would be helpful. Also, please specify the units on each map.

Thank you for this comment, we have adapted the figures as suggested.

Point-to-point response to reviewers' comments:

Reviewer #1 (Remarks to the Author):

Thanks for the careful revision from all the authors. After reviewing the responses, I am pleased to see that most of my comments have been addressed thoroughly. However, there are still two main areas that require further attention.

We thank the reviewer for acknowledging our careful revision. In the following, we will address the remaining concerns of the reviewer.

Regarding the abstract, one more sentence about the research gap or the significance of your research following the first sentence is necessary for the flow and logic of your abstract. Not only how important the grassland habitat is, but also you need to demonstrate how important is modeling the distribution of key grassland species under climate change.

We believe that we have already highlighted the importance of studying the changes in the distribution of key grassland species. The future grassland availability affects the mobility needs not only of pastoralist but also of wildlife, which are under a lot of pressure. Since we have already exceeded the allowed 150 words, we cannot further elaborate on this in the abstract. Nevertheless, we have rephrased the second sentence to stress the importance of modelling the distribution of key grass species.

Regarding the introduction, while I appreciate the efforts made to revise the introduction, I believe there is room for improvement in its structure and flow.

Specifically, the second paragraph illustrated the significance of the 7 key species in your study. It may come from the comments of other reviewers. However, I think you should generally demonstrate how important the key grassland species are to the whole grassland ecosystem, rather than focus solely on these 7 species. Some of the detailed explanations about the seven species could be moved to the methodology section or an appendix.

Thank you for pointing out this lack of information. We have added the following sentences to the introduction (lines 58-65):

“All the selected species are perennial plants and well adapted to arid and semi-arid ecosystems. The seven species occurred in about 60% of 659 plots of grasslands and dry savannah woodlands in East Africa⁸. Furthermore, these grasses represent the major structural inflorescence types of grasses: spikes and spike-like panicles, panicles and (digitate) racemes. The majority of the species have tufted and/or stoloniferous morphology and as such play an important role in the fire ecology, a key determinant of grasslands, soil hydrological properties and in providing much needed soil cover, facilitating water infiltration into the soil column and subsequently reducing run-off that causes soil erosion.”

Besides, the linking and flow of the introduction is not natural enough. You used a lot of stilted sentences and phrases, such as “The current state of research reveals (line 104)”, “The combination of these factors highlights a research gap that (line 112)”, “To address these data and knowledge gaps (line 115)”, etc. These phrases disrupt the

natural flow of the text. Consider rephrasing these sentences to create a more engaging and fluid narrative. Highlighting the research gap is important, but it should be done in a way that seamlessly draws the reader into the study.

We have rephrased several sentences of the introduction to obtain a more seamless narrative.

Finally, you spent too many lines illustrating your research objectives and contents (from line 115 to line 151) which, in my opinion, should be shorter. Even though, your research questions are still not clear. Hence, please clarify your research questions before illustrating your research objectives.

We have shortened this paragraph and emphasized the research questions again, woven into the text to obtain a seamless flow of the introduction.

From my side, the last paragraph of the introduction (from line 152 to line 158) is also redundant and unnecessary.

We have removed the last paragraph of the introduction.

Regarding your methodology, you explained why you modeled climatic data and the improvement of it compared to the existing databases. However, I still doubt the improvement of using high temporal resolution simulated data. The justification for using high temporal resolution simulated data remains unclear. Since the distribution and growth of grassland species are influenced by climatic conditions, rather than hourly weather conditions. A species will not present or disappear due to hourly, daily, or even monthly changing precipitation or temperature. Hence, in my opinion, you should at least elaborate on the benefits of using fine temporal predictors in SDMs or cite more relevant references to support your methodology.

I agree that a species will not disappear just because of hourly extremes, but the diurnal range for example or maximum and minimum temperatures play an important role. Since climate model outputs are mostly available on a monthly basis, the maximum and minimum temperatures are calculated based on monthly values, also the diurnal range is calculated based on monthly minimum and maximum temperature, which is by no means accurate. The hourly resolution allows to get a more realistic monthly minimum and maximum temperatures and therefore also a more accurate diurnal range, as used in the study (Bio 2). We agree that for now, it does not make sense to use fine temporal predictors for the SDMs, and therefore we also create monthly climatologies of the important bioclimatic variables. Nevertheless, they are based on high-resolution data, which makes the data more accurate. Also, the resolution in the horizontal and vertical space is important, particularly for precipitation. While temperature can be corrected with respect to elevation relatively well, this is not generally the case for precipitation. A study performed by one of the co-authors highlights the fact that global climate databases are not reliable in tropical mountains, particularly over Mt. Kilimanjaro (Hemp and Hemp, 2024). We think that we have addressed these points extensively in the method section and will therefore not further elaborate on this.

Once these issues are addressed, I believe that the manuscript will be ready for acceptance. Thank you for your continued efforts in improving the quality of the paper.

Thank you for your valuable and constructive feedback. Your suggestions have greatly contributed to improving the manuscript, and we hope that our revisions have fully addressed your final concerns.

Reviewer #2 (Remarks to the Author):

I am pleased to report that I am satisfied with the implemented changes. The revisions have effectively addressed all the concerns and suggestions previously highlighted. The improvements enhance the clarity, coherence, and overall quality of the work.

Thank you for your valuable and constructive feedback. Your suggestions have greatly contributed to improving the manuscript.

Reviewer #3 (Remarks to the Author):

Reading through the MS, the study has substantially improved. I only have a few comments.

Lines 127-133, it looks like unnecessary text here.

We have removed this part from the manuscript.

In the results, I would like to remove the quoted labels, such as "(Bio1)", "(A1)", "(B1)", and so on.

We thought that it would be helpful to have these labels to ease understandability, but if they are disturbing, we can remove them. We decided to leave them for the first appearance but removed them afterwards.

The conclusions are a bit farfetched and redundant, they can be condensed more.

We have condensed the conclusions as suggested.

Thank you for your valuable and constructive feedback. Your suggestions have greatly contributed to improving the manuscript, and we hope that our revisions have fully addressed your final concerns.